# LOCALLY ADAPTIVE CONFORMAL INFERENCE FOR OPERATOR MODELS

## ABSTRACT

Operator models are regression algorithms between Banach spaces of functions. They have become an increasingly critical tool for spatiotemporal forecasting and physics emulation, especially in high-stakes scenarios where robust, calibrated uncertainty quantification is required. We introduce Local Sliced Conformal Inference (LSCI), a distribution-free framework for generating function-valued, locally adaptive prediction sets for operator models. We prove finite-sample validity and derive a data-dependent upper bound on the coverage gap under local exchangeability. On synthetic Gaussian-process tasks and real applications (air quality monitoring, energy demand forecasting, and weather prediction), LSCI yields tighter sets with stronger adaptivity compared to conformal baselines. We also empirically demonstrate robustness against biased predictions and certain out-of-distribution noise regimes.

## 1 INTRODUCTION

An operator is a map $\Gamma : \mathcal{F} \mapsto \mathcal{G}$ between function spaces $\mathcal{F}$ and $\mathcal{G}$. Given a function $f \in \mathcal{F}$ as input, the operator returns another function $g = \Gamma(f)$. An operator model is a parameterized operator $\Gamma_\theta : \mathcal{F} \mapsto \mathcal{G}$ that is trained to predict functions $g \in \mathcal{G}$ given the function $f \in \mathcal{F}$. Analogous to ordinary regression, we learn the parameters $\theta \in \Theta$ by minimizing a function-valued loss $\mathcal{L} : \Theta \times (\mathcal{F}, \mathcal{G}) \mapsto \mathbb{R}$. Many scientific and engineering problems can be cast as operator learning problems, including partial differential equation (PDE) approximation Li et al. (2020); Sanderse et al. (2024), weather forecasting Pathak et al. (2022), climate downscaling Jiang et al. (2023), medical imaging (Maier et al., 2022), and image super-resolution Wei & Zhang (2023). In all of these problems, the output of interest is a curve or field, and uncertainty quantification (UQ) must therefore produce function-valued prediction sets rather than scalar intervals.

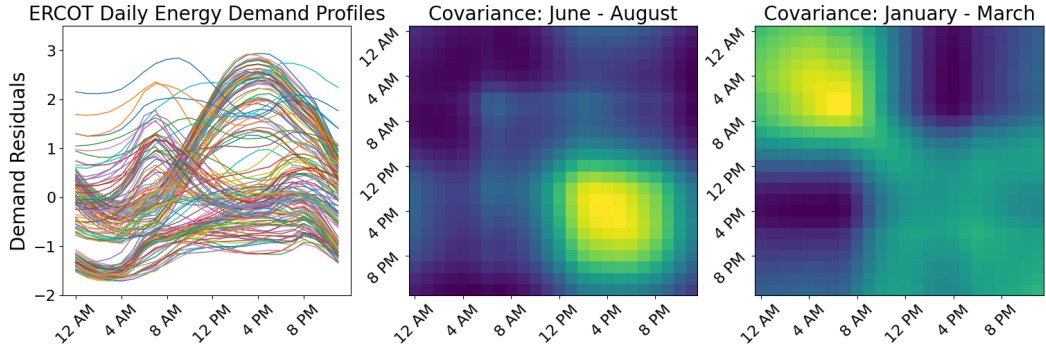

Figure 1: Residual functions from a neural operator model applied to energy demand (Section 4.2). (**Left**) Residuals vary smoothly across inputs, showing nonstationary amplitude and shape. (**Middle / Right**) Local residual covariance structures at two distant inputs. Their anisotropy and orientation differ substantially, illustrating the failure of global exchangeability and the need for geometrically adaptive local conformal methods.

One of the key challenges is that functional data in real systems are rarely stationary, identically distributed, or even exchangeable (Figure 1). In climate and environmental applications, residual

distributions drift gradually due to seasonal structure or long-term physical change; in power systems, load curves evolve across seasons (Figure 1); in operator-learning tasks, approximation errors vary smoothly with the structure and regularity of the input function. These settings violate the global exchangeability assumption underlying standard Conformal inference (CP) (Vovk et al., 2005; Shafer & Vovk, 2008). However, despite such global non-exchangeability, data are often locally stable: residual distributions associated with similar inputs tend to be similar, even when the global distribution changes. This motivates a local exchangeability framework (Campbell et al., 2019), where nearby inputs share approximately exchangeable residuals, but distant inputs do not.

Existing conformal methods designed for non-stationary or heterogeneous data typically introduce locality by modifying the quantile step through weighted order statistics or localized calibration rules (Barber et al., 2023; Guan, 2023). While effective in certain finite-dimensional problems, these approaches assume a global score geometry (e.g., a residual norm), which is often poorly suited to functional residuals whose uncertainty is anisotropic and structured due to continuity (Figure 1). Norm-based scores produce essentially spherical (isotropic) prediction regions in function space, regardless of the true shape of the residual variability.

Our contribution is to introduce local structure at the level of the conformity score itself rather than the quantile. We propose Local Sliced Conformal Inference (LSCI), a new, distribution-free method for constructing function-valued, locally adaptive prediction sets. LSCI builds a test-specific empirical residual distribution using similarity weights between the test input and calibration inputs. This localized residual distribution defines a depth-based conformity score, which measures the centrality of a residual relative to the specific local neighborhood of the test input. Unlike residual norms, depth functions adapt naturally to the shape of the local residual cloud, enabling prediction sets that deform correctly in regions of anisotropy, multimodality, or directional dependence.

## 2 BACKGROUND

We briefly review local exchangeability, conformal inference, and adaptive conformal inference to motivate our adaptive conformal inference approach for functional data. We denote $X \in \mathbb{R}^p$ as an input covariate vector, $Y \in \mathbb{R}^q$ as a target vector, and $f : \mathbb{R}^p \mapsto \mathbb{R}^q$ as a regression algorithm.

### 2.1 LOCAL EXCHANGEABILITY

The notion of local exchangeability, introduced by Campbell et al. (2019), relaxes global exchangeability by allowing distributions to evolve smoothly along an index set. Let $(Y_t)_{t \in T}$ be a stochastic process indexed by a set $T$ (e.g., time, space, or covariate index). The process is *exchangeable* if

$$(Y_t)_{t \in T} \overset{d}{=} (Y_{\pi(t)})_{t \in T} \tag{1}$$

for every injective map $\pi : T \to T$, i.e., every finite permutation of the index set. Exchangeability means that reordering the index set does not change the joint distribution.

Local exchangeability weakens this requirement by allowing small perturbations of the indexing to change the distribution in a controlled way. Let $d : T \times T \to [0, \infty)$ be a pre-metric on $T$ (not necessarily symmetric or satisfying the triangle inequality). Following Campbell et al. (2019), $(Y_t)_{t \in T}$ is said to be *locally exchangeable in* $(T, d)$ if, for every finite subset $A \subset T$ and every injective map $\pi : A \to T$,

$$d_{\text{TV}}\big(Y_A, Y_{\pi(A)}\big) \leq \sum_{t \in A} d\big(t, \pi(t)\big), \tag{2}$$

where $Y_A = (Y_t)_{t \in A}$, $Y_{\pi(A)} = (Y_{\pi(t)})_{t \in A}$, and $d_{\text{TV}}$ denotes total variation distance.

Condition (2) reduces to global exchangeability when $d \equiv 0$, and becomes vacuous when $d$ is unbounded. Intuitively, it says that configurations that are close in the index pre-metric must have similar joint distributions.

### 2.2 CONFORMAL INFERENCE

Let $\big\{(x_i, y_i)\big\}_{i=1}^n$ denote calibration data and let $\hat{f}$ be a predictor trained on an independent training set. A conformity score is any measurable function $S(x, y)$ that assigns larger values to less typical

pairs $(x, y)$ relative to the fitted model. A common choice in regression is a residual norm $S(x, y) = \|y - \hat{f}(x)\|_1$. We define the set of calibration scores as:

$$S_i = S(x_i, y_i), \qquad i = 1, \ldots, n,$$

and let $S_{(1)} \leq \cdots \leq S_{(n)}$ denote their order statistics. For a new input $x_{n+1}$, split conformal inference forms a prediction set

$$C(x_{n+1}) = \{y : S(x_{n+1}, y) \leq q_\alpha\},$$

where $q_\alpha = S_{(k)}$ is the $k = \lceil (n+1)(1-\alpha) \rceil$ largest value of the calibration scores. If the augmented sample $(x_1, y_1), \ldots, (x_n, y_n), (x_{n+1}, y_{n+1})$ is exchangeable (Eqn. 1), or more generally the scores $S_1, \ldots, S_n, S_{n+1}$ are exchangeable, then the standard symmetry argument of Vovk et al. (2005) yields finite-sample marginal validity,

$$\mathbb{P}\big(y_{n+1} \in C(x_{n+1})\big) \geq 1 - \alpha, \tag{3}$$

for any data distribution. In particular, (3) depends only on the exchangeability of the scores (or of the underlying data), and not on any correctness assumption on the model. Conversely, if the scores are not exchangeable, then (3) fails to hold.

## 2.3 Adaptive conformal inference

Several finite-dimensional extensions of conformal inference introduce locality by modifying the way the calibration quantile is computed Hore & Barber (2023); Barber et al. (2023); Guan (2023). Given a base score $S(x, y)$, these methods assign weights $w_i(x_{n+1})$ to the calibration points based on similarity between $x_i$ and the test input $x_{n+1}$, and then take a weighted empirical quantile of $\{S_i\}_{i=1}^n$ to define the threshold. Under suitable conditions, the resulting prediction sets achieve approximate coverage when the weighted empirical score distribution approximates the test-conditional score distribution.

The crucial point for our work is that such methods retain a global score function and introduce locality only through the quantile. In contrast, the method we propose localizes the score distribution itself while retaining the standard unweighted quantile rule. By localizing at the level of the conformity score rather than the quantile, LSCI is able to model the geometry of uncertainty in a way that standard conformal methods cannot, while preserving finite-sample guarantees and computational simplicity. Sections 3.2 and 3.3 develop this construction formally and analyze its coverage properties under local exchangeability of residuals.

## 3 Local sliced Conformal Inference

Let $\Gamma_\theta : \mathcal{F} \to \mathcal{G}$ denote an operator model. We assume $\mathcal{F}, \mathcal{G} \subset \mathcal{L}^2(\Omega)$, the space of square-integrable functions on a compact domain $\Omega \subset \mathbb{R}^p$ ($p \geq 1$). Let $\mathcal{D}_{\mathrm{tr}} = \{(f_s, g_s)\}_{s=1}^m$ be $m$ training pairs and $\mathcal{D}_{\mathrm{cal}} = \{(f_t, g_t)\}_{t=1}^n$ be $n$ calibration pairs; the indices $s$ and $t$ emphasize that these sets are disjoint. Let $\alpha \in (0, 1)$ be the miscoverage level, $f_{n+1}$ the test function, and $g_{n+1}$ its unknown target.

Our goal is to construct a prediction set $C_\alpha(f_{n+1}) \subset \mathcal{G}$ satisfying $\mathbb{P}(g_{n+1} \in C_\alpha(f_{n+1})) \geq 1 - \alpha$ and that is locally adaptive to heterogeneity in the conditional law of $g_{n+1} \mid f_{n+1}$ (e.g., changes in shape or variance). We assume the additive decomposition

$$g_t = \Gamma(f_t) + r_t, \tag{4}$$

where $\Gamma : \mathcal{F} \to \mathcal{G}$ is an unknown population operator and $(r_t)_{t \in \mathcal{T}}$ is a *locally exchangeable* error process (Campbell et al., 2019). Local exchangeability relaxes global exchangeability by allowing the distribution of $r_t$ to vary smoothly with $t$ (Section 2.1). This assumption allows for consistent local distribution estimation and retaining conformal guarantees (Section 3.3).

We denote $P_t \in \mathcal{P}(\mathcal{G})$ as the law of $r_t$ and $P \in \mathcal{P}(\mathcal{G})$ as the marginal mixture. Because $P_t$ is a distribution over functions, direct local empirical estimation is not possible as in standard univariate settings (Campbell et al., 2019; Guan, 2023; Hore & Barber, 2023). Instead, we use functional data depth (Liu, 1990; Zuo & Serfling, 2000) to characterize level sets of $P_t$ in function space. We first review $\Phi$-depths (Mosler & Polyakova, 2012), a functional depth family, and use them to define local $\Phi$-scores, which act as localized conformity measures on residuals $r_{n+1} = g_{n+1} - \Gamma_{\hat{\theta}}(f_{n+1})$. These scores induce "typical sets" that circumscribe the variability of $r_{n+1}$ at a given confidence level allowing us to define conformal inference sets $C_\alpha(f_{n+1})$.

### 3.1 Φ-DEPTH

**Data depth.** Data depth provides robust, order-based summaries (medians and quantile-like sets) of multivariate and functional distributions (Liu, 1990; Zuo & Serfling, 2000). For a function space $\mathcal{H} \subset \mathcal{L}^2(\Omega)$, element $h \in \mathcal{H}$, and probability measure $P \in \mathcal{P}(\mathcal{H})$, a depth function $d : \mathcal{H} \times \mathcal{P}(\mathcal{H}) \to [0, 1]$ quantifies the centrality of $h$ with respect to $P$ (0 = most outlying; 1 = most central). Common functional depths include integrated/infimum depths Mosler & Polyakova (2012); Mosler (2013), norm depths Zuo & Serfling (2000), band depths López-Pintado & Romo (2009), and shape-based depths Harris et al. (2021).

**Φ-depths.** Φ-depths (infimum depths) are a projection-based depth family that are robust and computationally efficient Mosler & Polyakova (2012), and, as we will see, easy to localize. Let $\Phi$ denote a family of continuous linear maps $\phi : \mathcal{H} \to \mathbb{R}^d$ (projections). Given a multivariate depth $D$ on $\mathbb{R}^d$ Zuo & Serfling (2000), define

$$D^{\Phi}(h \mid P) \;=\; \inf_{\phi \in \Phi} D\big(\phi(h) \,\big|\, \phi(P)\big), \tag{5}$$

where $\phi(P)$ is the pushforward of $P$ through $\phi$. We typically take $d = 1$ and use the univariate Tukey (half-space) depth $D(x \mid F) = 1 - |1 - 2F(x)| = 2\min\{F(x),\, 1 - F(x)\}$, with $F$ the (estimated) CDF of $\phi(h)$. Although any univariate depth will work (Section A.6), Tukey depths are straightforward to localize by re-weighting the empirical CDF. Φ-depths are non-degenerate in function spaces, affine-equivariant, robust to outliers, and decrease continuously from the center outwards Mosler & Polyakova (2012). Φ-depth, therefore, induce a proper center-out ordering from $D^{\Phi} = 1$ (most central) to $D^{\Phi} = 0$ (most outlying) on functional data sets.

The use of the projection family $\Phi$ can be interpreted geometrically as probing the function $h$ through a collection of one–dimensional linear functionals, each of which evaluates the curve from a different "direction." The depth $D_{\Phi}(h; P)$ therefore acts as a worst–case centrality measure: it records how typical $h$ appears under *every* projected view. This makes the score sensitive to anisotropy or non-spherical structure in the residual distribution that would be invisible to a scalar norm.

**Central regions.** Proper depth functions yield well-defined central regions of their target distribution $P$. For any $\gamma \in (0, 1)$, we define the $\gamma$-level central region of $P$ as

$$D_{\gamma}^{\Phi}(P) \;=\; \{\, h \in \mathcal{H} :\; D^{\Phi}(h \mid P) \geq \gamma \,\}. \tag{6}$$

Central regions are nested and expand monotonically as $\gamma \to 0$. Under standard regularity (e.g., uni-modality and convex level sets of $P$), the empirical versions converge to their population counterparts as the sample size grows. This means that central regions will often reflect the location, scale, and shape characteristics of $P$ Mosler & Polyakova (2012).

**Projection class.** The choice of $\Phi$ controls the slices used to probe $P$. Projection families may be fixed, data-driven, or random Mosler & Polyakova (2012). Fixed bases (e.g., Fourier, wavelets, splines) are efficient; data-driven projections such as functional principal components Ramsay & Dalzell (1991) yield compact representations. As illustrated in Table 4 (Appendix **??**), the slicing mechanism has little effect on marginal coverage. Thus, in general, we use normed Gaussian random slices as in the sliced Wasserstein distance (Bonneel et al., 2015).

### 3.2 METHOD

We model the conditional distribution of the response $g \in \mathcal{G}$ as $g \mid f \;=\; \Gamma(f) + r$, where the residual process $(r_t)_{t \in \mathcal{T}}$ is locally exchangeable (Section 2.1). Thus the new residual $r_{n+1}$ is locally exchangeable with the calibration residuals $r_1, \ldots, r_n$, which will allow us to evaluate the conformity of any $r \in \mathcal{G}$ with respect to the test-specific distribution $P_{n+1}$ through $\mathcal{D}_{\mathrm{cal}}$ and $f_{n+1}$.

**Local Φ-scoring.** We first train the operator $\Gamma_{\hat{\theta}}$ on $\mathcal{D}_{\mathrm{tr}}$. After training $\Gamma_{\hat{\theta}}$, we solely work with calibration residuals $r_t = g_t - \Gamma_{\hat{\theta}}(f_t)$ over $\mathcal{D}_{\mathrm{cal}}$ as in ordinary split conformal inference. Let $\Phi$ be a (uni/multivariate) linear projection family $\phi : \mathcal{G} \to \mathbb{R}^d$ and let $D$ be a depth on $\mathbb{R}^d$. Define the local Φ-score of $r$ at $f_{n+1}$ as the Φ-depth under $P_{n+1}$:

$$S^{\Phi}(r; P_{n+1}) \;:=\; D^{\Phi}(r \mid P_{n+1}) \;=\; \inf_{\phi \in \Phi} D\big(\phi(r) \,\big|\, \phi(P_{n+1})\big), \tag{7}$$

with $r = g - \Gamma_{\hat{\theta}}(f)$. For convenience, we take $d = 1$ and use univariate depths as in Section 3.1. Because we slice by $\phi \in \Phi$, computing $S^{\Phi}$ reduces to estimating univariate pushforwards $\phi(P_{n+1})$ for each $\phi \in \Phi$, rather than $P_{n+1}$ itself in function space. We estimate each projected (sliced) distribution $\phi(P_{n+1})$ by a locally weighted empirical measure:

$$\widehat{\phi}(P_{n+1}) \ = \ \sum_{t=1}^{n} w_t \, \delta(\hat{\phi}(r_t)) \ + \ w_{n+1} \, \delta(\infty), \quad w_t \geq 0, \quad \sum_{t=1}^{n+1} w_t = 1, \tag{8}$$

where $\delta(\infty)$ is at point mass at infinity representing the target function $g_{n+1}$. Weights are obtained from a similarity (localization) kernel $H : \mathcal{F} \times \mathcal{F} \to \mathbb{R}$ (Guan, 2023) centered at a statistical knockoff of the test feature, $\tilde{f}_{n+1} = f_{n+1} + \varepsilon, \ \varepsilon \sim \mathcal{GP}(0, K_{\sigma})$:

$$w_t \propto \exp\big( -\lambda H(f_t, \tilde{f}_{n+1})\big), \qquad w_{n+1} \propto \exp\big( -\lambda H(f_{n+1}, \tilde{f}_{n+1})\big). \tag{9}$$

Recent work (Hore & Barber, 2023) has shown that marginal coverage under local empirical measures can be guaranteed if we localize around statistical knockoffs of $f_{n+1}$, rather than $f_{n+1}$ directly. We will take $K_{\sigma}$ to be an identity kernel with variance $\sigma^2 = c^2 \, \mathrm{IQR}(f_t)^2$ and $c \in (0, 0.05)$. We also consider localizing feature maps $\varphi : \mathcal{F} \mapsto \mathcal{F}'$ and localizing on $H(\varphi(f_t), \varphi(f_{n+1}))$ (Chen et al., 2024) (Figure 2). Feature maps allow us to localize with respect to the underlying signal, or semantic content, of the inputs, rather than on their raw representation.

**Slice variance normalization.** Pooling projections $\{\phi_m(r_t)\}$ with different marginal scales, i.e., under heteroskedastic or locally-exchangeable data, can distort depths evaluations since most depths are only scale-equivariant (Mosler & Polyakova, 2012; Mosler, 2013). To ensure scale-invariance, we rescale each slice using the test-specific weights $w_t$ as:

$$s_m^2 \ = \ \frac{\sum_{t=1}^{n} w_t \, \phi_m(r_t)^2}{\sum_{t=1}^{n} w_t}, \qquad \widehat{\phi}_m(r_t) \ = \ \frac{\phi_m(r_t)}{s_m}, \quad \widehat{\phi}_m(r_{n+1}) \ = \ \frac{\phi_m(r_{n+1})}{s_m}.$$

Depths and quantiles are then computed on $\{\widehat{\phi}_m(r_t)\}$ and $\widehat{\phi}_m(r_{n+1})$. This preserves the per-slice depth-ordering of the calibration points while ensuring the slice statistics are locally scale-invariant.

**Local Conformal inference sets** To form the localized prediction set $C_{\alpha}(f_{n+1})$, we first compute each local calibration score $D^{\Phi}(r_t \mid P_{n+1})$ for $t = 1, \ldots, n$ using (7)–(8). Now, let $k = \lfloor \alpha(n+1) \rfloor$ and let $q_{\alpha}(f_{n+1})$ be the $k$-th smallest value among $\{D^{\Phi}(r_t \mid P_{n+1})\}_{t=1}^{n}$. The value $q_{\alpha}(f_{n+1})$ generates the test-specific residual central region

$$D_{\gamma(\alpha)}^{\Phi}(f_{n+1}) \ := \ \big\{ r \in \mathcal{G} : \ D^{\Phi}(r \mid P_{n+1}) \ \geq \ q_{\alpha}(f_{n+1}) \big\}, \tag{10}$$

and the conformal inference set is the prediction shifted region

$$C_{\alpha}(f_{n+1}) \ = \ \big\{ \Gamma_{\hat{\theta}}(f_{n+1}) + r : \ r \in D_{\gamma(\alpha)}^{\Phi}(f_{n+1}) \big\}, \tag{11}$$

as with Local conformal inference (LCP) Guan (2023) and Randomized LCP (RLCP) Hore & Barber (2023) prediction sets. Because the local $\Phi$-score is a positively oriented scoring rule (Section 2) the conformal inference regions are based on the $k = \lfloor \alpha(n+1) \rfloor$ order statistic. We denote $C_{\alpha}(f_{n+1})$ as a our Local Sliced Conformal Inference (LSCI) set.

Under full exchangeability, $C_{\alpha}(f_{n+1})$ attains exact marginal coverage while under local exchangeability, we provide an explicit finite-sample bound (Section 3.3). Empirically, the sets are highly robust to the choice of localizer $H$, localizing feature maps $\varphi : \mathcal{F} \mapsto \mathcal{F}'$, number of random slices $M$, bandwidth parameter $\lambda$, and depth function (Section 4.1 and Appendix 4). We again note that this score-localization mechanism leaves the conformal quantile step unchanged. We compute the standard unweighted conformal quantile of the localized scores, however, because the score distribution itself depends on the test input, the final threshold is still local and test-specific.

### 3.3 THEORY

As an uncertainty quantification (UQ) method, our goal is to guarantee coverage of the conformal inference sets to ensure their Frequentist validity. Our basic premise of local exchangeability, and even the act of localization in (8) (Guan, 2023), breaks exchangeability and thus invalidates the

standard conformal guarantee in (3). The coverage gap, however, can still be upper bounded (Barber et al., 2023) as

$$\mathbb{P}\big(g_{n+1} \in C_\alpha(f_{n+1})\big) \ \geq \ 1 - \alpha \ - \ \sum_{t=1}^{n} w_t \, d_{\mathrm{TV}}(R, R^t), \tag{12}$$

where $w_1, \ldots, w_n$ are the localization weights (Equation 9), $R = \{r_1, \ldots, r_{n+1}\}$ denotes all calibration residuals unioned with the test residual, $R^t$ is the set obtained by swapping $r_t$ and $r_{n+1}$, and $d_{\mathrm{TV}}(\cdot, \cdot)$ is total variation distance. Under full exchangeability, $d_{\mathrm{TV}}(R, R^t) = 0$ for all $t = 1, ..., n$, so coverage is exact. In the worst case $d_{\mathrm{TV}}(R, R^t) = 1$, rendering the bound vacuous. Thus, to obtain a useful guarantee we must quantify the degree of non-exchangeability.

**Proposition 3.1.** *Let $d : \mathcal{F} \times \mathcal{F} \to [0, 1]$ be a bounded pre-metric and suppose the residual process is locally exchangeable (Section 2.1). Then*

$$\Delta_n \equiv \sum_{t=1}^{n} w_t \, d_{\mathrm{TV}}(R, R^t) \ \leq \ \frac{1}{n+1} \sum_{t=1}^{n} d(f_t, f_{n+1}). \tag{13}$$

*Proof.* Deferred to Section A.3, though almost a direct consequence of local exchangeability (Eqn. (2)) applied to the residual process.

Under our local exchangeability assumption (Section 2.1), nearby covariates have approximately matching residual laws. Informally, if $f_t$ and $f_{n+1}$ are close in the pre–metric $d(\cdot, \cdot)$, then swapping $r_t$ and $r_{n+1}$ only slightly changes the joint distribution. This directly controls the coverage gap $\Delta_n$: when the residual process varies slowly over $\mathcal{F}$, permutations that move the test residual to nearby calibration indices incur only small total variation changes, so $\Delta_n$ is small and coverage remains close to the nominal level $1 - \alpha$. In the fully exchangeable special case, $d \equiv 0$ and the bound is exact.

Notice that (13) does not depend explicitly on the bandwidth $\lambda$ or the particular choice of localization kernel. In LSCI, $\lambda$ and the kernel H control the efficiency and shape of the prediction sets rather than the worst–case coverage bound. The effect of these choices is therefore investigated empirically in Section 4, where we show that coverage remains stable across a wide range of localization schemes, while the width and adaptivity of the bands vary in the expected way.

**Selecting the bandwidth.** The parameter $\lambda$ controls how the local empirical distribution $\widehat{P}_{n+1}$ concentrates around $f_{n+1}$ and how stable that estimate is. When $\lambda \to 0$, the weights become approximately uniform and LSCI reduces to a global depth–based conformal method: the approximate coverage gap $\Delta_n$ in (13) is then driven mainly by how far the residual process departs from global exchangeability. As $\lambda$ increases, the score distribution focuses on a smaller neighborhood of $f_{n+1}$, improving local adaptivity but also increasing variance in the empirical depth estimates. In practice we choose $\lambda$ on a disjoint tuning fold and compute the final threshold on a held–out calibration fold; Section 4 shows that coverage is empirically stable over a broad range of $\lambda$, while interval scores and band width are more sensitive.

### 3.4 SAMPLING THE PREDICTION SET

Depth-based prediction sets (Equation 11) are defined implicitly as subsets of the function space $\mathcal{G}$, which makes visualizing and apply them challenging. We therefore propose to approximate the set by drawing an ensemble of representative residual functions and shifting them by the point prediction (Harris & Sriver, 2024).

To visualize and summarize the implicit set $C_\alpha(f_{n+1})$, we draw residual samples in a localized FPCA basis fitted around $f_{n+1}$. We use inverse–transform sampling on the weighted projected coordinates and rejecting any sample whose depth falls below $q_\alpha(f_{n+1})$ (Algorithm 1, Appendix A.1). Pointwise empirical quantiles of the accepted samples, shifted by $\Gamma_{\hat{\theta}}(f_{n+1})$, produce the prediction bands used for IS and BW evaluation in Section 4. Full algorithmic details, including basis estimation and reconstruction, are provided in Appendix A.1. In general, we will sample two bands: one that satisfies expected coverage Mollaali et al. (2024); Moya et al. (2025) and one that satisfies high-probability coverage (coverage risk) Bates et al. (2021); Ma et al. (2024).

FPCA sampling is used only to visualize and summarize the conformal set. The conformal region is still defined analytically as in Section 3.2. This procedure generates representative samples from

this set to approximate its pointwise envelope when computing prediction metrics and for plotting (Section 4).

### 3.5 RELATED WORK

Standard split conformal methods use a single global threshold, which can be insensitive to heterogeneity across the feature space and thus yield overly conservative prediction sets. There exist many adaptive extensions, reviewed here, that attempt to address this by modifying the scoring rule or the calibration mechanism.

Local conformal methods adapt split conformal by weighting calibration examples via a similarity localizer and forming instance-wise sets from locally weighted CDFs/quantiles Guan (2023); Barber et al. (2023); Hore & Barber (2023); RLCP further uses knockoff localization to retain marginal validity Hore & Barber (2023). LSCI extends LCP/RLCP to operator models by replacing vector scores with depth-based functional local $\Phi$-scores to calibrate directly in function space. Alternative *adaptive scoring* methods rescale residuals using an auxiliary variance model $\hat{\sigma}(\cdot)$, but reusing training data can understate uncertainty and harm coverage (Romano et al., 2019); we compare to functional variants Diquigiovanni et al. (2022); Lei et al. (2015); Moya et al. (2025). *Conformalized quantile regression* (CQR) fits conditional quantiles and then conformalizes (Romano et al., 2019), but may struggle at extreme quantiles and produce wide sets Guan (2023); we include functional CQR-like baselines (Ma et al., 2024; Angelopoulos et al., 2022; Mollaali et al., 2024).

Beyond conformal UQ, probabilistic and Bayesian neural operators, such as last-layer Laplace (Magnani et al., 2022), Bayesian DeepONets Garg & Chakraborty (2022); Zhang et al. (2023), linearized operators as GPs Magnani et al. (2024), and probabilistic NOs with proper scoring rules (Bülte et al., 2025), provide predictive uncertainty. However, they do not carry the same distribution-free, finite-sample guarantees that LSCI and other conformal methods offer.

## 4 EXPERIMENTS

We evaluate three synthetic GP-based tasks: (i) 1D regression, (ii) 1D autoregressive forecasting, and (iii) 2D spherical autoregressive forecasting. Details of each data-generating mechanism are provided in Appendix A.4. For all tasks we use a four-layer, 64-channel Fourier Neural Operator (FNO) Li et al. (2020); we set 16 Fourier modes for the 1D problems and $16 \times 32$ modes for the 2D problem.

We report the following metics: Functional coverage (FC), Expected coverage (EC), Coverage Risk (CR), Band Width (BW), and Interval Score (IS). FC is the probability that the whole field is covered, EC and CR are the average and high-probability coverage across space, respectively, BW reflects the overall size of the band, and IS is a strictly proper scoring for interval forecasts. FC, EC, and CR represent three different notions of coverage considered in the functional conformal literature. Our approach guarantees FC, while our sampler allows us to generate ensembles approximately meeting EC (Mollaali et al., 2024; Moya et al., 2025) or CR (Ma et al., 2024). BW and IS reflect the overall precision of the prediction band, where smaller numbers mean more precise uncertainty. Exact computational details provided in the appendix A.4.

### 4.1 SYNTHETIC EXPERIMENTS

We first verify that marginal coverage (Proposition 3.1) holds across a range of localizer kernels $H$, localizing feature maps $\varphi(\cdot)$ number of slicing directions $N$, and kernel bandwidths $\lambda$. We include exchangeable and locally exchangeable settings to measure the empirical coverage gap (Eqn. (12)).

We will consider three localizing kernels: an $L_\infty$-Norm localizer $d_{\inf}(f_1, f_2) = \exp(-\lambda \|\varphi(f_1) - \varphi(f_2)\|_\infty)$, an $L^2$-Norm localizer $d_2(f_1, f_2) = \exp(-\lambda \|\varphi(f_1) - \varphi(f_2)\|_2)$ and k-nearest neighbor localizer $d_{\mathrm{knn}}(f_1, f_2)$, which is the $L^2$ localizer considering only the nearest $k$ neighbors. We include four feature maps: the identity function $\varphi(f) = f$, a truncated functional PCA projection (32 components), a truncated Fourier projection (16 modes), and the learned operator embedding $\varphi(f) = \Gamma_{\hat{\theta}}(f)$. We use $\lambda = 0.5, 1, 2$ and approximate the local $\Phi$-scores using $N = 1, 10, 100, 200$ slice projections. Each combination is applied to the same exchangeable 1D Gaussian process regression task (Appendix A.4) and the coverage is estimated over 50 simulation replicates.

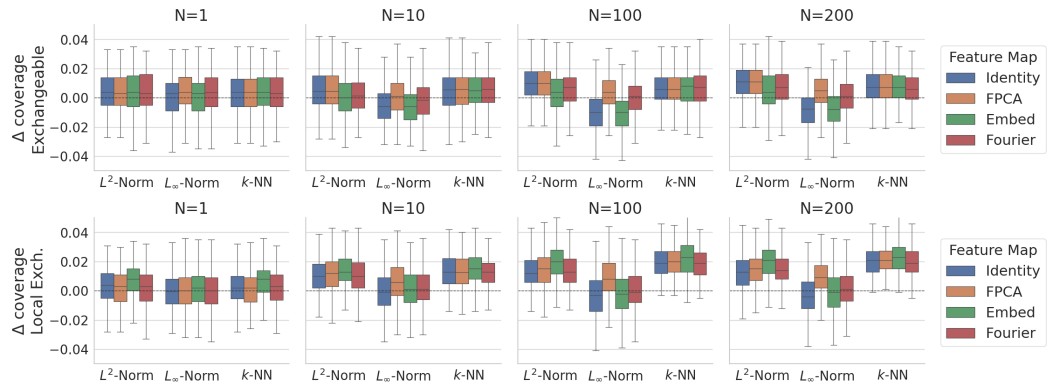

Figure 2: LSCI empirical coverage ($\alpha = 0.1$) on homoskedastic regression across many $H$–$\varphi$ and $\lambda$-$M$ localization settings. Coverage in either case not strongly impacted by localization.

Figure 2 shows near-nominal coverage ($\alpha = 0.1$) across all $H$–$\varphi$ and $\lambda$-$M$ combinations for LSCI. Increasing $M$ can slightly de-stabilize coverage, due to numerical instabilities estimating the infimum in the local $\Phi$-scores (7) when there are many ties (e.g. $d_{\inf}(\cdot)$). Using soft-minimum Boyd & Vandenberghe (2004) stabilizes coverage, but induces a slight upward bias ($\approx 0.005 - 0.01$). Overall, the empirical coverage gap predicted by Proposition 3.1 for LSCI appears small. Coverage is evaluated under the true conformal sets, not the samples (Alg. 1).

In practice, we recommend choosing the localization kernel $H$ and bandwidth $\lambda$ by cross-validation on the calibration residuals, selecting the pair that minimizes empirical interval score. When the targets are high-dimensional (high resolution), using a low-dimensional feature map $\varphi(\cdot)$ (e.g., FPCA, Fourier modes, or learned embeddings) stabilizes distances; when functions are smooth then the identity feature map is sufficient. Figure 2 illustrate that LSCI coverage is robust to these choices. In practice, we found Fourier feature maps and $L_\infty$-norm localizers to consistently perform well.

**Baseline comparisons** We compare LSCI to the conformal baselines on the three different heteroskedastic GP tasks (Appendix A.4). **Reg-GP1D** is univariate GP regression with global variance changes, **AR-GP1D** is AR(1) univariate GP forecasting with spectral variance changes, and **AR-GP2D** is AR(1) bivariate GP forecasting with local variance changes. In all cases, we train, calibrate, and test on 1000 samples per split. Reg-GP1D and AR-GP1D results averaged over 25 simulation replicates, AR-GP2D results only averaged over 5 due to computational cost.

Table 1: Coverage and interval metrics on Gaussian-process simulations. Coverage (either FC, EC, or CR) should be high (up to $0.9$), while interval score (IS) should be low.

| Method | Reg-GP1D – Global Het. | | | | AR-GP1D – Spectral Het. | | | | AR-GP2D – Local Het. | | | |
|---|---|---|---|---|---|---|---|---|---|---|---|---|
| | FC ↑ | EC ↑ | CR ↑ | IS ↓ | FC ↑ | EC ↑ | CR ↑ | IS ↓ | FC ↑ | EC ↑ | CR ↑ | IS ↓ |
| *Baselines* | | | | | | | | | | | | |
| Conf. | 0.900 | 0.999 | 0.999 | 3.779 | 0.888 | 0.998 | 0.996 | 2.380 | 0.914 | 0.942 | 0.976 | 1.900 |
| Supr. | 0.902 | 0.993 | 0.980 | 2.706 | 0.891 | 0.995 | 0.991 | 2.152 | 0.890 | 1.000 | 1.000 | 3.020 |
| UQNO | 0.776 | 0.973 | 0.903 | 1.691 | 0.561 | 0.969 | 0.892 | 1.512 | 0.000 | 0.940 | 0.912 | 1.734 |
| PONet | 0.527 | 0.901 | 0.683 | 1.363 | 0.206 | 0.897 | 0.587 | 1.496 | 0.000 | 0.901 | 0.542 | 1.839 |
| QONet | 0.516 | 0.917 | 0.689 | 1.360 | 0.134 | 0.898 | 0.567 | 1.467 | 0.000 | 0.906 | 0.582 | 1.852 |
| *Proposed* | | | | | | | | | | | | |
| LSCI1 | 0.909 | 0.975 | 0.901 | 1.935 | 0.904 | 0.966 | 0.885 | 1.430 | 0.972 | 0.979 | 0.976 | 0.892 |
| LSCI2 | 0.912 | 0.973 | 0.893 | 1.609 | 0.906 | 0.976 | 0.933 | 1.442 | 0.916 | 0.996 | 0.998 | 1.444 |
| LSCI3 | 0.909 | 0.904 | 0.655 | 1.200 | 0.904 | 0.899 | 0.586 | 0.997 | 0.972 | 0.948 | 0.862 | 0.786 |
| LSCI4 | 0.912 | 0.900 | 0.629 | 1.026 | 0.906 | 0.909 | 0.605 | 0.984 | 0.916 | 0.983 | 0.976 | 1.160 |

Conformal baselines include low-rank functional sets with Gaussian scoring Lei et al. (2015) (Conf); conformalized integrated band method (Supr) Diquigiovanni et al. (2022); conformalized probabilistic

deep-operator model (PONet) and its quantile-regression variant (QONet) Moya et al. (2025) We also include the calibrated UQ for neural operators approach (UQNO) Ma et al. (2024). All methods are tuned on the calibration data to achieve their respective conformal guarantees. Deep Operators Nets (QONet, PONet) trained separately using their prescribed MLP architectures. We do not include Bayesian neural-operator baselines because their UQ depends heavily on prior/architectural choices and does not yield distribution-free, finite-sample guarantees.

For LSCI, we include two variants: one using $L_\infty$-Norm localization and one using $k$-NN localization ($k = 500$). The former uses Fourier feature maps and the latter uses identity feature maps for localization. For each setting, we sample one band with approximately $\alpha = 0.1$ EC to compare against QONet and PONet, and one with approximate $\alpha = 0.1$ CR to compare with UQNO. This gives us four combinations LSCI1 ($L_\infty$-Norm, $\alpha = 0.1$ CR), LSCI2 ($k$-NN, $\alpha = 0.1$ CR), LSCI3 ($L_\infty$-Norm, $\alpha = 0.1$ EC), LSCI4 ($k$-NN, $\alpha = 0.1$ EC). We enforce the desired guarantee by adjusting the pointwise empirical quantiles of the accepted samples.

Table 1 shows that the sampled LSCI sets achieve strong coverage and risk control across all synthetic tasks. In particular, if we compare within methods that control FC (Conf, Supr, LSCI), we see that LSCI consistently has lower IS. Similarly, for methods that control the coverage risk (CR) (UQNO) or EC (PONet, QONet), the corresponding LSCI sets, at that level, tend to have lower interval scores. This indicates that the LSCI are, potentially, better adapting to the heterogeneity, rather than simply expanding their widths. However, this affect is modulated by the chosen localizer $H$. For global heterogenity, $k$-NN localizers had lower IS, while for local heterogenity $L_\infty$ localizers had lower IS. Thus, the localizer can also impact the efficiency the prediction sets, which is primarily driven by the depth function. Depth-based scores shrink their sets along directions where the local residual cloud is tight and only expand it along directions of high variability because they are based on central regions (Section 3.1). Thus, for the same nominal coverage we avoid wasting width in low-variance directions and thereby reduce interval scores.

**How many samples?** Table 2 shows LSCI's empirical performance is only mildly dependent on the number of samples. Re-using the **AR-GP1D** setting from Table 1, we see that as long as the EC is controlled at the same level, the interval scores do not vary much with increasing $n_s$. Thus, small conformal samples can be sufficient for practical application.

Table 2: IS doesn't vary with increasing sample sizes $n_s$ as long as EC is controlled.

| $n_s =$ | 50 | 500 | 1000 | 2000 | 5000 |
|---|---|---|---|---|---|
| EC | 0.922 | 0.932 | 0.933 | 0.932 | 0.929 |
| IS | 1.024 | 1.038 | 1.042 | 1.040 | 1.024 |

Computationally, drawing $n_s$ samples and evaluating the local depth score scales linearly in both the calibration size and $n_s$. For the dataset sizes considered here this cost is modest, and for larger calibration sets one can combine LSCI with approximate nearest-neighbor search in feature space without affecting coverage.

**Biased predictors & covariate shift** Finally, we evaluate each method when the predictor is biased and when the data experiences covariate shift over time (from train to calibration to test) (Shimodaira, 2000). These represent realistic scenarios, particularly where operator models are often applied (e.g. environmental and physical processes).

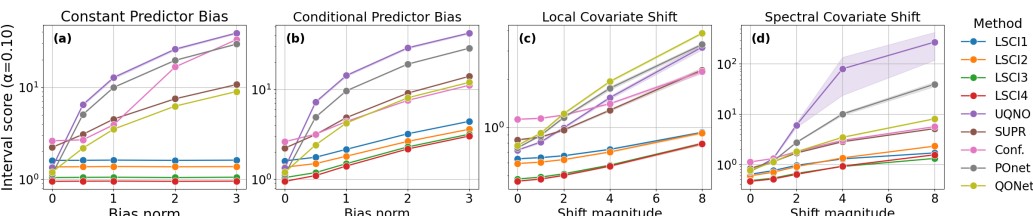

Figure 3: **a**. Constant bias $2\sin(4\pi t)$, re-normed to the given bias level, added to each prediction. **b**. Conditional bias $2c\|f\|_2 \sin(4\pi t + \|f\|_2)$. **c**. Local covariate shift via a moving $\sigma$ "bump" (Section A.4). **d**. Spectral covariate shift via a rotating $\sigma$ "spike" through the harmonics of $f$ (Section A.4)

Figure 3(a) and 3(b) show that LSCI's interval score (IS) is unaffected by fixed biases and increases more slowly than alternative methods when the bias depends on the input process. Figure 3(c)

and 3(d), show that LSCI also robust against certain kinds of covariate shift and out-of-distribution behavior. Many baseline methods try to counteract covariate shift and predictor bias by expanding their intervals, hence their increasing interval scores. Simulation details in (Section A.4).

## 4.2 EXPERIMENTS ON REAL DATA

We evaluate LSCI against the baseline methods on three real-world tasks. **Air Quality:** Daily PM2.5 profiles from a site in Beijing, China constructed from hourly measurements (UCI dataset 501). **Energy Demand:** Daily 24-hour energy demand curves from the Electric Reliability Council of Texas (ERCOT); constructed from hourly measurements (eia.gov/electricity). **Weather-ERA5:** global 2-meter surface temperature on a $32 \times 64$ latitude–longitude grid, aggregated into daily averages (Hersbach et al., 2020). Energy Demand and Weather-ERA5 are lag 1 forecasting tasks, Air Quality predicts PM2.5 profiles from concurrent temperature, precipitation, and dew point profiles.

Table 3: Uncertainty metrics for all conformal methods applied to energy forecasting, weather forecasting, and air quality prediction.

| | Energy Demand | | | | Air Quality | | | | Weather-ERA5 | | | |
|---|---|---|---|---|---|---|---|---|---|---|---|---|
| Method | FC $\uparrow$ | EC $\uparrow$ | BW $\uparrow$ | IS $\downarrow$ | FC $\uparrow$ | EC $\uparrow$ | BW $\uparrow$ | IS $\downarrow$ | FC $\uparrow$ | EC $\uparrow$ | BW $\uparrow$ | IS $\downarrow$ |
| *Baselines* | | | | | | | | | | | | |
| Conf. | 0.582 | 0.981 | 2.135 | 2.217 | 0.883 | 0.989 | 1.845 | 2.851 | 0.950 | 0.876 | 6.681 | 8.327 |
| Supr. | 0.633 | 0.939 | 1.396 | 1.646 | 0.000 | 0.879 | 0.479 | 2.096 | 0.876 | 1.000 | 18.08 | 18.09 |
| UQNO | 0.513 | 0.913 | 1.353 | 1.690 | 0.000 | 0.161 | 0.091 | 3.851 | 0.000 | 0.916 | 4.572 | 5.654 |
| PONet | 0.496 | 0.841 | 0.895 | 1.466 | 0.565 | 0.894 | 203.7 | 232.5 | 0.000 | 0.889 | 15.63 | 21.23 |
| QONet | 0.482 | 0.802 | 1.016 | 1.759 | 0.000 | 0.296 | 15.68 | 217.3 | 0.000 | 0.890 | 13.293 | 16.65 |
| *Proposed* | | | | | | | | | | | | |
| LSCI1 | 0.892 | 0.935 | 1.518 | 1.546 | 0.887 | 0.676 | 0.243 | 0.433 | 0.919 | 0.990 | 5.362 | 5.418 |
| LSCI2 | 0.909 | 0.934 | 1.513 | 1.540 | 0.937 | 0.967 | 0.731 | 0.839 | 0.957 | 0.994 | 5.608 | 5.631 |
| LSCI3 | 0.892 | 0.897 | 1.227 | 1.257 | 0.887 | 0.659 | 0.229 | 0.424 | 0.919 | 0.985 | 4.836 | 4.916 |
| LSCI4 | 0.909 | 0.897 | 1.216 | 1.257 | 0.937 | 0.917 | 0.479 | 0.599 | 0.957 | 0.991 | 5.152 | 5.187 |

Table 3 shows UQ metrics for the three datasets. In all cases, LSCI yields valid prediction sets (FC $\approx 0.9$) with good expected coverage on either band (EC $\approx 0.9$). LSCI's bands are competitive with or tighter than baselines with correspondingly lower interval scores, indicating a high degree of adaptivity. For operator-learning applications, these functional bands provide uncertainty quantification at the level of entire solution fields, allowing practitioners to assess whether a predicted field is globally reliable, to identify regions where uncertainty is concentrated, and to compare models in a way that aligns with downstream physical tasks. In particular, Weather-ERA5 shows that LSCI can strongly improve over non-adaptive methods. Additional results in Appendix A.5 show LSCI detecting seasonal cycles (Charlton-Perez et al., 2024; Beverley et al., 2024; Mouatadid et al., 2023) in Weather-ERA5 that the underlying neural operator misses.

## 5 DISCUSSION

We introduced Local Sliced Conformal Inference (LSCI), a framework for function-valued, locally adaptive prediction sets for operator models. By combining projection-based $\Phi$-depths with knockoff-localized conformal calibration, LSCI captures structured residual variability while retaining distribution-free guarantees. Across synthetic and real tasks, LSCI yields tighter, more adaptive sets than conformal baselines (Sections 4.1–4.2) The main limitation is computational overhead from localization and sampling at test time; batching, caching projections, and parallel/GPU evaluation help mitigate but do not eliminate this non-insignificant cost. Sampling from high resolution fields could become prohibitively expensive. Furthermore, if there are abrupt, rather than smooth, changes in the conditional distribution, then local kernels will oversmooth these regions leading to poor adaptivity around sudden breaks or changepoints. Future work includes structured multivariate outputs (e.g., multi-level temperature fields) via multi-channel projections, changepoint adaptivity, learned localizers, and faster sampling mechanisms.

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

# A APPENDIX / SUPPLEMENTAL MATERIAL

## A.1 SAMPLING ALGORITHM

Our sampler works in locally adapted functional principal component (FPCA) coordinates (Ramsay & Dalzell, 1991). We (i) estimate a local FPCA basis around the test feature, (ii) sample projected coordinates by inverse transform from the weighted empirical pushforwards $\phi_k(P_{n+1})$, and (iii) reconstruct candidate residuals and accept them if they lie in the local depth region.

---

**Algorithm 1** LSCI residual sampling (inverse transform)

---

**Input:** $D^{\Phi}_{\gamma(\alpha)}(f_{n+1})$, $\Gamma_{\hat{\theta}}$, $\Phi$, $H$, $M$ (projections), $n_s$ (samples)

1. Sample knockoff $\tilde{f}_{n+1} = f_{n+1} + \varepsilon$, $\varepsilon \sim \mathcal{GP}(0, K_\sigma)$; Compute and normalize weights $w_t \leftarrow \mathrm{softmax}\big(H(f_t, \tilde{f}_{n+1})\big)$ for $t = 1, ..., n+1$.

2. Compute calibration residuals $r_t = g_t - \Gamma_{\hat{\theta}}(f_t)$; weighted mean $\bar{r}_{n+1} = \sum_t w_t r_t$; and first $M$ weighted eigenfunctions $\{\psi_k\}_{k=1}^M$.

3. For $k = 1{:}M$, set $\xi_{t,k} = \langle r_t - \bar{r}_{n+1}, \psi_k \rangle$ and form spectral CDFs $\widehat{F}_k(x) = \sum_t w_t \mathbf{1}\{\xi_{t,k} \leq x\}$.

**Sampling loop** (until $n_s$ accepted)
    1. Draw $u_k \sim U(0,1)$, set $\tilde{\xi}_k = \widehat{F}_k^{-1}(u_k)$ for $k = 1{:}M$.
    2. Reconstruct $\tilde{r} = \bar{r}_{n+1} + \sum_{k=1}^M \tilde{\xi}_k \psi_k$.
    3. **Accept** if $D^{\Phi}(\tilde{r} \mid P_{n+1}) \geq q_\alpha(f_{n+1})$; else resample.

**Return:** $\{\tilde{r}_{n+1}^{\,i}\}_{i=1}^{n_s}$ and $\tilde{C}_\alpha(f_{n+1}) = \{\Gamma_{\hat{\theta}}(f_{n+1}) + \tilde{r}_{n+1}^{\,i}\}_{i=1}^{n_s}$.

---

The inverse-transform step treats the projected coordinates $\phi_k(r)$ as independent for proposal generation. The final depth-based rejection step helps correct this approximation and ensures samples lie inside the local central region. Increasing $M$ improves expressivity but may reduce the acceptance rate. In practice, however we observe close to 100% acceptance with $M = 32$ and $M = 128$ components on 1D and 2D regression tasks.

## A.2 LOCAL EXCHANGEABILITY

Let $(Y_t)_{t \in \mathcal{T}}$ denote a stochastic process on $\mathbb{R}$ with finite first and second moments. $(Y_t)_{t \in \mathcal{T}}$ is exchangeable if

$$(Y_t)_{t \in \mathcal{T}} =_D (Y_t)_{\pi(t) \in \mathcal{T}},$$

for all injective maps $\pi : \mathcal{T} \mapsto \mathcal{T}$, i.e., for all permutations of the indexing set Campbell et al. (2019). Exchangeability means that re-ordering $(Y_t)_{t \in \mathcal{T}}$ along $\mathcal{T}$ does not change its distribution. Exchangeability is ordinarily required to prove the finite-sample validity of conformal inference sets, which are based on the adjusted quantiles of the empirical measure.

Local exchangeability is a recent generalization of exchangeability that assumes $(Y_t)_{t \in \mathcal{T}}$ is not exchangeable, but that elements close in the indexing set are close to exchangeable. $(Y_t)_{t \in \mathcal{T}}$ is locally exchangeable in $\mathcal{T}$ if for any subset $T \subset \mathcal{T}$ and injective map $\pi : T \mapsto \mathcal{T}$

$$d_{TV}(Y_T, Y_{\pi(T)}) \leq \sum_{t \in T} d(t, \pi(t)) \tag{14}$$

where $Y_T$ is $(Y_t)_{t \in \mathcal{T}}$ restricted to $T$, $Y_{\pi(T)}$ is $(Y_t)_{t \in \mathcal{T}}$ restricted to $\pi(T)$, $d_{TV}$ is the total variation distance (Sason & Verdú, 2016), and $d : \mathcal{T} \mapsto \mathcal{T}$ is a pre-metric on $\mathcal{T}$.

Local exchangeability is critical because, while each $Y_\tau$, $\tau \in \mathcal{T}$, follows its own distribution $G_\tau$, we can approximate $G_\tau$ with a local empirical measure

$$\hat{G}_\tau = \sum_{t \in T} \eta_t(\tau) \delta(Y_t) \tag{15}$$

where $\delta(Y_t)$ is a Dirac point mass at $Y_t$ and $\eta_t(\tau)$ are localization weights. These weights are defined as

$$\eta_t(\tau) = \max\{0, M_\tau^{-1} + 2(\mu_\tau - d(t, \tau))\}$$

$$M_\tau = \max_M \left\{ \left( M^{-1} \sum_{t=1}^{M}(1 + 2d_m(\tau)) \right) \geq 2d_M(\tau) \right\}, \quad \mu_\tau = M_\tau^{-1} \sum_{t=1}^{M_\tau} d(t, \tau) \tag{16}$$

where $m, M \in 1, ..., T$ and $d_m(\tau)$ is the $m$'th smallest distance. Thus, we can use the adjusted quantiles of the local empirical measure to construct a local conformal inference set for each $Y_t \in (Y_t)_{t \in \mathcal{T}}$.

## A.3 PROOFS

Proof of proposition 3.1. Let $d : \mathcal{F} \times \mathcal{F} \mapsto \mathbb{R}$ is a bounded pre-metric on $\mathcal{F}$. Without loss of generality we assume $0 \leq d(f_1, f_2) \leq 1$. The coverage gap in proposition 3.1 and equation 12 is defined as

$$\sum_{t=1}^{n} w_t d_{TV}(R, R^t), \tag{17}$$

where $R = \{r_t = g_t - \Gamma_{\hat{\theta}}(f_t) : (f_t, g_t) \in \mathcal{D}_{cal}\}$ and $R^t$ is the set $R$ under the permutation function $\pi : T \mapsto T$, which swaps $r_t$ with $r_{n+1}$, and leaves all other elements unchanged. By the definition of local exchangeability (Definition 14), and our use of un-weighted quantiles:

$$\sum_{t=1}^{n} w_t d_{TV}(R, R^t) \leq \frac{1}{n+1} \sum_{t=1}^{n} \sum_{i=1}^{n} d(f_i, f_{\pi(i)}). \tag{18}$$

However, because $d(f_i, f_{\pi(i)}) = 0$ for all $i \neq t$ since $i = \pi(i)$ in this case, the upper bound reduces to $\sum_{t=1}^{n} w_t d(f_t, f_{n+1})$. For notational convenience, we let $d_t = d(f_t, f_{n+1})$ and write

$$\sum_{t=1}^{n} w_t \sum_{i=1}^{n} d(f_i, f_{\pi(i)}) = \frac{1}{n+1} \sum_{t=1}^{n} d_t. \tag{19}$$

## A.4 SIMULATION DETAILS

**Metrics** We report the following metics. Let $B_i(u) = [L_i(u), U_i(u)]$ denote a prediction band on the grid and $g_i$ a target function observed on the grid $u_j \in \mathcal{U}$. Let $c_i = p^{-1} \sum_{j=1}^{p} \mathbf{1}(g_i(u_j) \in B_i(u_j))$. We define functional coverage (FC) as $\text{FC} = m^{-1} \sum_{i=1}^{m} \mathbf{1}(c_i = 1)$, the expected coverage (EC) $\text{EC} = m^{-1} \sum_{i=1}^{m} c_i$ Mollaali et al. (2024); Moya et al. (2025), and the coverage risk (CR) $\text{CR}_{0.1} = m^{-1} \sum_{i=1}^{m} \mathbf{1}(c_i \geq 1 - 0.1)$ (Bates et al., 2021; Ma et al., 2024). We also include the interval

score, IS $= m^{-1} \sum_{i=1}^{m} p^{-1} \sum_{j=1}^{p} [(U_i(u_j) - L_i(u_j)) + (2/\alpha)(L_i(u_j) - g_i(u_j))_+ + (2/\alpha)(g_i(u_j) - U_i(u_j))_+]$ a strictly proper scoring rule for interval forecasts Gneiting & Raftery (2007) to measure band quality. Finally, we measure the band width (BW) $\mathrm{BW} = m^{-1} \sum_{i=1}^{m} p^{-1} \sum_{j=1}^{p} U_i(u_j) - L_i(u_j)$ to measure precision.

**Data** We generate the Gaussian process data in Section 4.1 as follows.

**Experiment 0: 1D Homoskedastic GP (Figure 2)** We generate three independent splits (train, calibration, test), each with $n_{\text{train}} = n_{\text{cal}} = n_{\text{test}} = 1000$ functional pairs $\{(f_t, g_t)\}$ on a 1D grid $\mathcal{U} = \{u_i\}_{i=1}^{128} \subset [0, 1]$ of $p = 128$ equispaced points. At each discrete time $t \in \{1, \ldots, 1000\}$ we draw Gaussian–process innovations

$$\varepsilon_t^f \sim \mathcal{GP}(0, K_f), \qquad\qquad \varepsilon_t^g \sim \mathcal{GP}(0, K_g),$$

independent across $t$ and between processes. The data are then formed as

$$f_t(u) = \sigma_f \, \varepsilon_t^f(u),$$
$$g_t(u) = 0.6 \, f_t(u) + \sigma_g \, \varepsilon_t^g(u), \qquad u \in \mathcal{U},$$

with *constant* scales $\sigma_f = 0.35$ and $\sigma_g = 0.25$ (no heteroskedasticity). For each process we use an RBF kernel with jitter,

$$K_f(u, v) = \exp\left(-\frac{(u-v)^2}{2\,\ell_f^2}\right) + \lambda\,\mathbf{1}\{u = v\}, \qquad K_g(u, v) = \exp\left(-\frac{(u-v)^2}{2\,\ell_g^2}\right) + \lambda\,\mathbf{1}\{u = v\},$$

with length-scales $\ell_f = 0.15$, $\ell_g = 0.08$ and jitter $\lambda = 10^{-3}$. The three splits (train, calibration, test) are generated independently under this specification on the shared grid $u \in \mathcal{U} \subset [0, 1]$.

**Experiment 1: 1D Global–Heteroskedastic GP (i.i.d.). (Table 1)** We generate three independent splits (train, calibration, test), each with $n_{\text{train}} = n_{\text{cal}} = n_{\text{test}} = 1000$ functional pairs $\{(f_t, g_t)\}$ on a 1D grid: $\mathcal{U} = \{u_i\}_{i=1}^{128} \subset [0, 1]$ of $p = 128$ equispaced points. As in the previous experiment, at each discrete time $t \in \{1, \ldots, 1000\}$ we draw Gaussian–process innovations

$$\varepsilon_t^f \sim \mathcal{GP}(0, K_f), \qquad\qquad \varepsilon_t^g \sim \mathcal{GP}(0, K_g),$$

independent across $t$ and between processes. The data are then formed as

$$f_t(u) = \sigma_t^f(u)\, \varepsilon_t^f(u),$$
$$g_t(u) = 0.6 \, f_t(u) + \sigma_t^g(u)\, \varepsilon_t^g(u).$$

For each process we use an RBF kernel with jitter,

$$K_f(u, v) = \exp\left(-\frac{(u-v)^2}{2\,\ell_f^2}\right) + \lambda\,\mathbf{1}\{u = v\}, \qquad K_g(u, v) = \exp\left(-\frac{(u-v)^2}{2\,\ell_g^2}\right) + \lambda\,\mathbf{1}\{u = v\},$$

with length-scales $\ell_f = 0.15$, $\ell_g = 0.08$ and jitter $\lambda = 10^{-3}$. Both processes use time-varying but spatially constant scales ("global" heteroskedasticity),

$$\sigma_t^f(u) \equiv \sigma_f \, g_f(t), \qquad \sigma_t^g(u) \equiv \sigma_g \, g_g(t),$$

with base levels $\sigma_f = 0.35$, $\sigma_g = 0.25$. The functions $g_f(t)$ and $g_g(t)$ are smooth sinusoidal ramps in $t$ normalized to have mean 1, producing mild temporal modulation of variance while preserving independence across time (no autoregression). The three splits (train, calibration, test) are generated independently under the same specification on the shared grid $u \in \mathcal{U} \subset [0, 1]$.

**Experiment 2: 1D Spectral Heteroskedastic GPs (AR(1)) (Table 1)** We generate three independent splits (train, calibration, test), each with $n_{\text{train}} = n_{\text{cal}} = n_{\text{test}} = 1000$ functional pairs $\{(f_t, g_t)\}$ on a 1D grid $\mathcal{U} = \{u_i\}_{i=1}^{128} \subset [0, 1]$ of $p = 128$ equispaced points. The dynamics follow a lagged–response scheme $f_{t+1}(u) \equiv g_t(u)$, initialized by a GP draw for $f_0$. We set

$$f_0(u) = \sigma_f \, \varepsilon_0^f(u), \qquad \varepsilon_0^f \sim \mathcal{GP}(0, K_f),$$

where $K_f$ and $K_g$ are radial basis function (RBF) kernels with jitter,

$$K_f(u,v) = \exp\left(-\frac{(u-v)^2}{2\,\ell_f^2}\right) + \lambda\,1\{u=v\}, \qquad K_g(u,v) = \exp\left(-\frac{(u-v)^2}{2\,\ell_g^2}\right) + \lambda\,1\{u=v\},$$

with length-scales $\ell_f = 0.02$ (used in the $f_0$ initialization), $\ell_g = 0.08$ (used for $g$–innovations), and jitter $\lambda = 10^{-6}$. For $t \geq 1$, we draw GP innovations $\varepsilon_t^g \sim \mathcal{GP}(0, K_g)$ and form AR(1) residual fields

$$R_t^g(u) \;=\; \rho\,R_{t-1}^g(u) \;+\; \sqrt{1-\rho^2}\,\varepsilon_t^g(u), \qquad \rho = 0.9, \quad R_0^g \equiv 0,$$

so that the residual variance is time–stationary. We set a linear mean linkage

$$\mu_t(u) \;=\; 0.6\,f_t(u),$$

and define

$$g_t(u) \;=\; \mu_t(u) \;+\; \sigma_t^g(u)\,R_t^g(u).$$

The latent driver then updates $f_{t+1}(u) \equiv g_t(u)$. The scale field for $g_t$ varies across space via a low–frequency Fourier expansion with $H = 2$ harmonics,

$$\sigma_t^g(u) \;=\; \sigma_g\Big[1 + \sum_{k=1}^{2} a_{k,t}\,\phi_k(u)\Big],$$

where $\{\phi_k\}$ are sinusoidal basis functions on $[0,1]$. The coefficients are *linked* to the current driver $f_t$ via projections,

$$a_{k,t} \;\propto\; \langle f_t, \phi_k \rangle \;=\; \int_0^1 f_t(u)\,\phi_k(u)\,du,$$

with a mean–preserving normalization so that $\int \sigma_t^g(u)\,du = \sigma_g$ (fixed base level). We use base scales $\sigma_f = 0.35$ (appearing only in the initialization of $f_0$) and $\sigma_g = 0.40$.

Using $f_{t+1} \equiv g_t$ with $g_t(u) = 0.6\,f_t(u) + \sigma_t^g(u)R_t^g(u)$, yields temporally coupled fields with AR(1) residual dynamics in $g$ and spatially structured, $f_t$–linked spectral heteroskedasticity in the variance of $g_t$. Train, calibration, and test splits are generated independently.

**Experiment 3: 2D Local Heteroskedastic GP (Table 1)**  We generate three independent splits (train, calibration, test), each with $n_{\text{train}} = n_{\text{cal}} = n_{\text{test}} = 1000$ functional pairs $\{(f_t, g_t)\}$ on a 2D grid

$$\mathcal{U} = \{(u_1^{(i)}, u_2^{(j)})\}_{i=1,\ldots,32;\,j=1,\ldots,64} \subset [0,1]^2$$

of $p_1 = 32$, $p_2 = 64$ equispaced points. At each discrete time $t \in \{1, \ldots, 1000\}$ we draw spatial Gaussian–process innovations

$$\varepsilon_t^f \sim \mathcal{GP}(0, K_f), \qquad \varepsilon_t^g \sim \mathcal{GP}(0, K_g),$$

independent across $t$ and between processes. Let $\tau_t = \sin(2\pi t/T)$ be a scalar temporal trend with $T = 1000$. The fields are formed as

$$f_t(u) \;=\; \sigma_t^f(u)\,\varepsilon_t^f(u) \;+\; \tau_t, \qquad g_t(u) \;=\; 0.6\,f_t(u) \;+\; \sigma_t^g(u)\,\varepsilon_t^g(u) \;+\; \tau_{t+1},$$

for $u \in \mathcal{U}$. Thus $g_t$ includes a one–step lead of the trend relative to $f_t$. There is no temporal autoregression (i.i.d. over $t$ conditional on the scales). For each process we use a separable 2D RBF kernel with jitter,

$$K_f\big((u_1, u_2), (v_1, v_2)\big) = \exp\left(-\frac{(u_1-v_1)^2}{2\,\ell_f^2}\right)\exp\left(-\frac{(u_2-v_2)^2}{2\,\ell_f^2}\right) \;+\; \lambda\,1\{(u_1, u_2) = (v_1, v_2)\},$$

$$K_g\big((u_1, u_2), (v_1, v_2)\big) = \exp\left(-\frac{(u_1-v_1)^2}{2\,\ell_g^2}\right)\exp\left(-\frac{(u_2-v_2)^2}{2\,\ell_g^2}\right) \;+\; \lambda\,1\{(u_1, u_2) = (v_1, v_2)\},$$

with isotropic length-scales $\ell_f = 0.15$ and $\ell_g = 0.08$ and jitter $\lambda = 10^{-6}$.

Both processes use time–varying *local* scales of the form

$$\sigma_t^f(u) \;=\; \sigma_f\Big[1 + \alpha_f\,\kappa\Big(\frac{u-c(t)}{w}\Big)\Big], \qquad \sigma_t^g(u) \;=\; \sigma_g\Big[1 + \alpha_g\,\kappa\Big(\frac{u-c(t)}{w}\Big)\Big],$$

where $\sigma_f = 0.35$, $\sigma_g = 0.40$ are base levels, $\alpha_f, \alpha_g > 0$ set the contrast, $w = (0.06, 0.06)$ is the (axis–wise) width, and $\kappa$ is a smooth, nonnegative bump function (e.g., Gaussian) centered at $c(t)$. The center $c(t) \in [0,1]^2$ traces a circular path over time, so the region of elevated variance moves smoothly across the domain. Under this specification, $\{(f_t, g_t)\}$ are temporally independent given the local scale fields, with $g_t$ combining a linear response to $f_t$, spatial GP noise at scale $\sigma_t^g(u)$, and a one–step–ahead temporal trend. Train, calibration, and test splits are generated independently.

**Experiment 4: Constant prediction bias (Figure 3a)**  This setup mirrors *Experiment 1* except for two changes: (i) *spectral* heteroskedasticity replaces the global heteroskedasticity for both processes and (ii) we inject an evaluation (predictor) bias into $g_t$ in the calibration/test splits only:

$$\tilde{g}_t(u) = g_t(u) + b(u), \qquad b(u) = c\sin(4\pi u),$$

with $b(u)$ RMS–normalized to amplitude $c > 0$. The training split remains unbiased. Each split contains $n = 1000$ pairs generated independently under this specification. This allows us to arbitrarily bias the calibration/test target functions away from the training functions.

**Experiment 5: Conditional prediction bias (Figure 3b)**  This experiment is identical to *Experiment 4*, except that the evaluation bias added to $g_t$ in the calibration/test splits now depends on the current covariate $f_t$. Define the RMS of $f_t$ over the grid

$$\phi_t = \left(\tfrac{1}{p}\sum_{i=1}^{p} f_t(u_i)^2\right)^{1/2}, \quad p = 128,$$

and set an amplitude $A_t = 2\,\phi_t$. For $u \in [0,1]$ we introduce a phase-shifted sinusoidal bias

$$b_t(u) = A_t\,\sin\!\big(4\pi u + \phi_t\big),$$

then normalize its root-mean-square (RMS) to a prescribed level $c > 0$:

$$\tilde{b}_t(u) = \frac{c}{\left(\tfrac{1}{p}\sum_{i=1}^{p} b_t(u_i)^2\right)^{1/2}}\,b_t(u).$$

The calibration and test observations are thus reported as

$$\tilde{g}_t(u) = g_t(u) + \tilde{b}_t(u),$$

while the training split remains unbiased (no $b_t$ added). Each split contains $n = 1000$ pairs generated independently under this specification.

**Experiment 6: Local covariate shift (Figure 3c)**  We generate a single trajectory $\{(f_t, g_t)\}_{t=1}^{3000}$ on the 1D grid $\mathcal{U} = \{u_i\}_{i=1}^{128} \subset [0,1]$ under the same data–generating mechanism as *Experiment 2* except using *local* heteroskedasticity. We then form contiguous splits: $\mathcal{T}_{\text{train}} = \{1,\dots,1000\}$, $\mathcal{T}_{\text{cal}} = \{1001,\dots,2000\}$, $\mathcal{T}_{\text{test}} = \{2001,\dots,3000\}$. The local scale fields for both processes evolve over time with a *linear* ramp in amplitude and a moving spatial "bump," so the marginal distribution of the covariates drifts across the trajectory. Consequently, $P_{\text{train}}(f) \neq P_{\text{cal}}(f) \neq P_{\text{test}}(f)$, i.e., the three splits differ systematically in the input distribution (earlier times have smaller variance and a different high–variance location than later times). The conditional mechanism is unchanged: the mean mapping $g_t(u) \mid f_t$ remains $0.6\,f_t(u)$ (with the same heteroskedastic noise structure), so this constitutes *covariate shift* induced purely by the temporal partitioning of a nonstationary process.

**Experiment 7: Spectral Covariate Shift (Figure 3d)**  This mirrors *Experiment 6* but replaces *local* heteroskedasticity with the *spectral* heteroskedasticity scheme defined earlier. The scale fields $\sigma_t^f(u)$ and $\sigma_t^g(u)$ are expanded in low–frequency Fourier modes with a *linear* ramp in amplitude over time, inducing nonstationary variance. As a result, the covariate shift across splits arises from changing *spectral* content, i.e., time–varying weights on low–frequency modes, rather than a moving spatial bump.

## A.5 SPATIAL ADAPTIVITY

Figure 4 shows the generated upper and lower 90% LSCI band *on the residual process* (Equation 10) across four seasons of the Weather-ERA5 data. The bands clearly exhibit spatially varying seasonality, with the northern and southern hemispheres accurately oscillating throughout the year. Thus, the bands are able to account for seasonal variations that the base FNO model was not able to represent. These patterns are consistent with well-documented seasonally dependent biases in both dynamical and machine-learning forecast systems, which motivate local calibration rather than a single global threshold (Charlton-Perez et al., 2024; Beverley et al., 2024; Mouatadid et al., 2023).

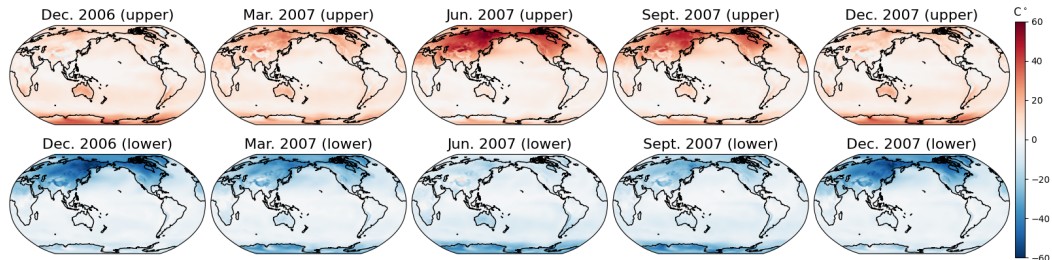

Figure 4: Spatial uncertainty as a function of seasonality. LSCI adapts over time to the seasonal patterns.

### A.6 INVARIANCE TO DEPTH AND LOCALIZER

Finally, we verify that the marginal-coverage guarantee (Proposition 3.1) holds across a range of projection families $\Phi$, depth notions $D$, localizers $H$, and kernel bandwidths. Although alternative depth notions and projection schemes are not considered in this manuscript, they could just as well replace the proposed Tukey depth and Gaussian random slices.

**Different $\Phi$ projectors.** We consider the following projection families: randomized slice sampling (Rand), Functional Principal Components (FPCA), a wavelet basis (Wave), FPCA with randomized slices (R-FPCA), and wavelets with randomized slices (R-Wave). For the univariate depth $D$ in (5), we include Tukey depth, $\ell_\infty$ depth, and Mahalanobis depth, representing Type A, B, and C constructions, respectively Zuo & Serfling (2000). Each method is applied to a 1D Gaussian-process regression task with heteroskedastic variance, and marginal coverage is estimated over 100 simulation replicates.

|  | Tukey | $\ell_\infty$ | Mahal. |
|---|---|---|---|
| Rand | 0.902 $\pm 0.02$ | 0.905 $\pm 0.01$ | 0.904 $\pm 0.02$ |
| FPCA | 0.902 $\pm 0.01$ | 0.906 $\pm 0.01$ | 0.908 $\pm 0.01$ |
| Wave | 0.905 $\pm 0.01$ | 0.903 $\pm 0.01$ | 0.902 $\pm 0.01$ |
| R-FPCA | 0.901 $\pm 0.02$ | 0.901 $\pm 0.02$ | 0.901 $\pm 0.01$ |
| R-Wave | 0.904 $\pm 0.02$ | 0.905 $\pm 0.02$ | 0.903 $\pm 0.02$ |

Table 4: Coverage ($\alpha = 0.1$) by depth $D$ and projection family $\Phi$ with $2\sigma$ error bars.

Table 4 shows near-nominal coverage ($\alpha = 0.1$) across all $\Phi$–$D$ combinations. Adding randomization to data-driven (FPCA) or fixed (Wave) bases yields slight improvements. Overall, the empirical coverage gap predicted by Proposition 3.1 appears small. Coverage here is evaluated under the true LSCI conformal sets, not the samples (Alg. 1).

**Different $H$ localizers.** We next evaluate LSCI under three localizers: an $\ell_2$ kernel $H(f_t, f_s) = \exp(-\lambda\|f_t - f_s\|_2)$, an $\ell_\infty$ kernel $H(f_t, f_s) = \exp(-\lambda\|f_t - f_s\|_\infty)$, and a $k$-nearest-neighbor kernel with $k = (1 + \lambda)^{-1}n$ (rounded to an integer). The bandwidth $\lambda \geq 0$ controls localization strength. Table 5 shows nominal *marginal* coverage across localizers and bandwidths. Coverage is, again, evaluated under the true conformal sets not the samples (Alg. 1).

|  | $\ell_2$ | $\ell_\infty$ | k-NN |
|---|---|---|---|
| $\lambda = 1$ | 0.903 $\pm 0.02$ | 0.903 $\pm 0.01$ | 0.905 $\pm 0.01$ |
| $\lambda = 2$ | 0.904 $\pm 0.02$ | 0.904 $\pm 0.02$ | 0.902 $\pm 0.02$ |
| $\lambda = 3$ | 0.904 $\pm 0.02$ | 0.905 $\pm 0.02$ | 0.904 $\pm 0.01$ |
| $\lambda = 4$ | 0.904 $\pm 0.02$ | 0.904 $\pm 0.02$ | 0.903 $\pm 0.02$ |
| $\lambda = 5$ | 0.904 $\pm 0.02$ | 0.903 $\pm 0.02$ | 0.903 $\pm 0.02$ |

Table 5: Coverage ($\alpha = 0.1$) by localizer $H$ and bandwidth $\lambda$ with $2\sigma$ error bars.

