# OpenReview forum: "Locally adaptive conformal inference for operator models"
_ICLR.cc/2026/Conference — Submitted to ICLR 2026_

### Official Review · Reviewer_NpHH · 2025-10-29

**Soundness:** 3
**Presentation:** 2
**Contribution:** 3
**Rating:** 6
**Confidence:** 2

**Summary:**

This paper introduces Local Sliced Conformal Inference (LSCI), a CP algorithm for neural operator models that constructing locally adaptive, function-valued prediction sets. The method cleverly uses $\Phi$-depth functions (inf of a family of linear maps) for functional conformity scores and similarity-localized calibration to generate prediction sets that adapt to heterogeneous residual distributions. The authors provide finite-sample validity guarantees under local exchangeability and demonstrate improvements over existing conformal baselines on synthetic and real-world datasets.

**Strengths:**

I have to admit that the operator model exposition is a little beyond me, so I'll be commenting mostly from a conformal prediction perspective.

- It is a nice contribution to extend CP to operator learning and translate both the algorithm and guarantees to functional space. The authors have showed that this extension is nontrivial and yields better intervals than other (learning based) uncertainty quantification methods.

- Other than good direct experiment results, I find the ablation study / robustness analysis in Figures 1 and 2 and Tables 2,4,5 to be thoughtful and convincing. The authors demonstrated that coverage is stable across various choices of localizers, feature maps, projection families, and depth functions (since CP should be model-agnostic), and show meaningful advantages over baselines in biased prediction / distribution shift scenarios.

- On theory, the authors drew the connection between the localized calibration bound (Eq 13) to the tradeoff between localization and calibration data availability (Eq 14). Although more clarity can be desired (for example through experiments), the guidance is helpful for readers and practitioners.

**Weaknesses:**

I found the paper to be a bit difficult to follow due to my lack of background in operator learning. (might not be a weakness).

For example, I didn't understand the significance of the statistical knockoff, why the Tukey (half-space) depth was selected, how exactly $\lambda$ is tuned to balance the trade-off, and how FPCA sampling recreates the conformal interval.  Although the authors did try to explain these choices/algorithms, the current explanations are either rushed or a little hand-wavy and could benefit from more principled explanations, through equations and examples and plots, utilizing space in the appendix.

Another question that I had after reading this paper is, how is this UQ useful for Operator learning? The authors introduced 4 metrics, but in my experience did not explain what are the implications of each thoroughly. (I think FC and IS are intuitive for me, but how should I interpret the other two in the context of operator learning?) Maybe it is obvious for the operator learning community, but more likely it's the case that neither community knows what to do with this nice method you created. Some discussion on the properties and usefulness of the prediction sets, that needs to be created from rejection sampling and then empirical quantiles.

**Questions:**

See weaknesses.

---

> ### Author Response · Authors · 2025-12-04
> **Rebuttal by authors**
>
> We appreciate your thoughtful review and are glad that the ablation, robustness analysis, and theoretical components were helpful from a conformal prediction perspective. We have revised the manuscript to make the operator-learning background, hyperparameter choices, and interpretation of the evaluation metrics clearer.
>
> __W1__:  In the revised version, we trimmed the neural-operator exposition in the main text and moved several implementation details to the appendix. We now explain more directly what matters for LSCI and avoid most of the previous neural operator discussion. Whats most import is that the model outputs are functions, the residuals are functional objects, and these residuals often vary smoothly with the input function. Our intention is that a reader without operator-learning expertise can still follow the full conformal pipeline.
>
> __W2__: We agree that this cluster of design choices was likely not sufficiently motivated in the first version. The revised manuscript attempts to clarify each of these points.
>  - For the statistical knockoff the key here is that you cant guarantee coverage at the exact point of localization, but you can for any point _near_ the point of localization. By perturbing the input we make it so that our point of interest is now a nearby point of out point of localization. Thus coverage can be guaranteed.
>  - The choice of Tukey (half-space) depth is fairly arbitrary since nearly any non-parametric univariate depth will give almost the exact same results (Appendix A6.). The Tukey depth is a very common choice in the depth literature and was convenient to localize since its based on an empirical CDF. All we had to do was reweight the CDF and the tukey depth would be automatically localized.
> - The role of $\lambda$ is now made more concrete in Section 3.3, where we describe how it governs the tradeoff between stability and locality, and how we tune it in practice using the interval score on a held-out fold.
>  - The FPCA-based sampling procedure is now described more cleanly and we are clear to note that FPCA is only used to generate a finite ensemble of residual functions from this set for visualization and for computing band-based metrics. Concretely, we fit a localized FPCA basis around $f_{n+1}$, sample projected coordinates with inverse-transform sampling under the localized weights, reject any sample whose depth falls below $q_\alpha(f_{n+1})$, and shift accepted residuals by $\Gamma_\theta(f_{n+1})$.. Pointwise empirical quantiles of this ensemble form the plotted bands and are used for IS and BW. We now say this explicitly in Section 3.4 and move the full algorithmic details to Appendix A.1.
>
> __W3__:  We agree that the original text did not do enough to connect the four metrics back to operator-learning tasks. In the revision, Sections 4 and A.4 now spell out the role of each:
>  - FC (functional coverage): fraction of test functions whose entire trajectory lies inside the band. For operator learning, this measures whether the learned operator, coupled with LSCI, can reliably “contain” full function-valued trajectories (e.g., full spatial fields or full daily demand curves).
>  - EC (expected coverage): average fraction of time/space points covered. This is directly relevant when partial coverage is acceptable as long as most of the function is captured, which is common in climate and energy applications where pointwise errors are tolerated as long as gross structure is correct.
>  - CR (coverage risk): high-probability guarantee that coverage is at least $1 - \delta$ on most test functions. This is a tail-type metric used in recent CP work.
>  - BW and IS: BW measures the average width of the band; IS is a strictly proper scoring rule for interval forecasts that penalizes both width and miscoverage. In operator learning, these quantify how much extra “slack” the UQ layer adds on top of the deterministic operator. Smaller BW/IS at fixed FC/EC/CR means the operator + LSCI UQ  is providing tighter, more informative uncertainty around the learned functional mapping.
>
> We also now make clear that CR, EC, BW, and IS are computed from the sampled approximation, while FC is computed from the analytic depth level set. This distinction clarifies how to use the method in practice and how to interpret the metrics across different operator-learning tasks.
>
> We thank you again for your constructive feedback and for giving us the opportunity to improve the clarity and accessibility of the presentation.

---

### Official Review · Reviewer_8P52 · 2025-10-30

**Soundness:** 2
**Presentation:** 2
**Contribution:** 2
**Rating:** 2
**Confidence:** 3

**Summary:**

In this paper, the authors propose a method for constructing prediction sets for functional data. Using the tool of $\Phi$-depths, the authors establish local $\Phi$-scores, which act as localized conformity measures on residuals. Furthermore, by employing this score along with conformal prediction methods, the authors construct prediction sets for functional data. The authors also provide a more intuitive form of the constructed prediction sets through sampling. Finally, the authors present relevant theoretical properties and experimental results. The experimental results demonstrate that the proposed method outperforms baseline methods in terms of both coverage rate and the size of the prediction sets.

**Strengths:**

The authors employ the tool of $\Phi$-depths to establish conformity measures suitable for functional data. Additionally, the authors utilize local information to construct the score function, thereby endowing the proposed method with enhanced robustness.

**Weaknesses:**

1. For methodology, in the literature on localized prediction sets (e.g., Guan, 2023[1]; Hore & Barber, 2023[2]; Barber, 2023[3]), the threshold $q_{\alpha}(f_{n+1})$ is typically defined as the $1-\alpha$ quantile of a weighted distribution $\sum_{i=1}^{n+1}w_i\delta_i$, where the weight $w_i$ assigned to each data point reflects its contribution to the construction of the prediction set. However, in line 201, the authors adopt a different quantile definition. Constructing prediction sets in this way may substantially undermine the validity of the proposed approach.

2. Theoretically, in line 223, the theoretical result presented by the authors is not supported by the literature Barber (2023)[3], as the underlying methodologies differ. Moreover, the authors do not provide a detailed proof for it.

**Questions:**

1 In this paper, local information is used to construct the $\Phi$-scores, yet no local information is utilized when calculating threshold $q_{\alpha}(f_{n+1})$. This differs from all existing conformal prediction frameworks (e.g., Guan, 2023[1]; Hore & Barber, 2023[2]; Barber, 2023[3]). Why not use existing localized conformal prediction methods to compute threshold $q_{\alpha}(f_{n+1})$? Is the currently used threshold a reasonable one?

2 The theoretical results presented in Line 223 are referenced to Barber (2023) [3]. However, there is a methodological discrepancy between the two works, particularly in the calculation of the threshold. Given this difference, the results from Barber (2023) [3] cannot directly support the authors' claim. It is hoped that the authors can provide a corresponding explanation or present the specific proof process.

3 Depth-based prediction sets are defined implicitly as subsets of the function space. Therefore, I am curious about how the metrics in the experimental section were calculated, particularly the band width (BW)?

---

> ### Author Response · Authors · 2025-12-04
> **Rebuttal by authors**
>
> We thank you for your comments and for highlighting several points of confusion. We have substantially revised the manuscript to clarify the methodology, the theoretical result, and the computation of metrics.
>
> __W1__: You are correct that our method differs from prior localized conformal approaches in where locality is introduced. Existing methods localize the quantile via weighted order statistics, whereas LSCI localizes the score and then applies a standard (unweighted) conformal quantile.
>
> In the revised manuscript, Sections 2 and 3 now make this distinction explicit and derive the appropriate coverage bound for this setting. We still cite Barber’s general inequality, which is appropriate here, since it is written in terms of local weights and a deviation from exchangeability term. LSCI corresponds to the flat-weight special case (since we do not localize the quantile), and our local exchangeability assumption directly plays the role of the deviation term. We now include the short, self-contained proof in Appendix A.2.
>
> __W2__: Thank you for pointing out the inconsistency in the earlier bound. The original submission mistakenly reused the local weights from the score within the quantile derivation, which unintentionally implied a doubly localized scheme. We considered this method during development but did not implement it.
>
> In the revision, we correct this by returning to the original form of Barber’s bound expressed via local weights and a deviation from exchangeability term. We now specialize this inequality to the flat-weight case for LSCI and substitute our local exchangeability criterion to simplify it. This gives a simpler and correct coverage-gap bound that matches the actual algorithm we used. The revised appendix now clearly delineates which steps follow Barber (2023) and which arise from our score-localization construction.
>
> __Q1__: Localizing the score contours the residual distribution in a data-dependent way that allows our prediction sets to adapt their location and geometry to the local residual distribution. This allows the depth-based approach to construct highly efficient prediction sets by shrinking and expanding anisotropically along different directions of variability. Once the score has been localized, applying the standard conformal quantile ensures approximate marginal coverage under local exchangeability. Weighting the quantile again (i.e., “double localization”) is therefore not necessary. By comparison, using only a single, fixed, global score induces a fixed geometry on the prediction sets (e.g. a sphere for L2 distances) and weighting the quantile scales its radius in all directions. This can result in far less efficient prediction sets. We see this phenomenon empirically via the lower Interval Scores in local exchangeability settings for the baseline methods.
>
> __Q2__: As noted above, the original mismatch arose because we incorrectly reused the local weights in the derivation, effectively implying a doubly localized scheme that _we did not implement_. In the revision, we explicitly separate these two components and have fully corrected our bound. We apologize for this oversight.
> - We now state clearly that our method corresponds to the flat-weight case in Barber’s general bound.
> - We derive the coverage-gap bound by starting from their inequality and then substituting our local exchangeability condition.
>
> __Q3__: This was indeed not clear in the original version. LSCI defines a prediction set as a depth-based level set in function space, which is implicit. In the revised manuscript, we now explain how each metric is computed:
>
> Functional coverage (FC) is evaluated analytically with respect to the depth-defined set, using the conformal residuals and the explicit depth threshold.
>
> Coverage risk (CR), expected coverage (EC), band width (BW), and interval score (IS) are computed from a sampled ensemble of functions drawn from the depth thresholded region. To compute these, we sampled two ensembles, one that attained 90% EC coverage and one that attained 90% CR coverage. This was done to make them comparable to existing methods that control EC or CR, rather than FC. In other words, we use sampling as a way to obtain a finite, discrete representation of the implicit functional set, and we then compute CR, EC, BW, and IS on this ensemble exactly as one would in a finite-dimensional setting. We now make this distinction explicit: FC is tied directly to the analytic set, while the other metrics are defined via the sampled approximation, because those quantities are not naturally defined for arbitrary infinite-dimensional sets without such a discretization.

---

### Official Review · Reviewer_ngnT · 2025-10-31

**Soundness:** 4
**Presentation:** 1
**Contribution:** 2
**Rating:** 2
**Confidence:** 5

**Summary:**

The paper proposes a locally adaptive conformal score for models that map between function spaces. The present a depth-based conformal score function using a localized empirical cumulative distribution function with weights determined by a similarity kernel. They provide provable upper bounds on the coverage gap obtained from breaking the global exchangeability assumption. They validate their method on synthetic data and real-world data (Air Quality, Energy Demand, and Weather data).

**Strengths:**

The authors validate their method with an intuitive upper bound for the coverage gap, which makes it clear how coverage suffers when local exchangeability is weakened. The experiments are quite robust and further strengthen their proposed method.

**Weaknesses:**

Main Weaknesses
* The proposed method isn’t well-motivated. The paper didn’t cite any examples where global exchangeability might break or local exchangeability might hold with functional data.
* It’s not easy to see why depth-based scores are important to obtain local marginal coverage or tight prediction sets. In experiments, it’s clear that LSCI outperforms all the conformal baselines in the Interval Score metric, but there is no intuition behind why depth-based score can reduce Interval scores.
* There doesn’t seem to be any experiments validating the coverage gap bound in Proposition 3, which appears to be key result.

Writing-related weaknesses
* The conformal prediction background section needs to cover local exchangeability and local adaptive conformal inference in the finite-dimensional setting.
* The introduction has a lot of unnecessary background on neural operators and fails to motivate the problem well.

**Questions:**

* Can a depth-based score in a finite-dimensional setting outperform conformal baselines?

---

> ### Author Response · Authors · 2025-12-04
> **Rebuttal by authors**
>
> We thank you for your thoughtful comments and questions. Below we respond to each point and reference the corresponding revisions made in the current manuscript.
>
> __W1__: We agree that the earlier version did not do enough to motivate when local exchangeability is a reasonable assumption for functional data. In the revised manuscript, the Introduction and Section 2 have been reworked to foreground this. In the revised text, we now:
>      - Use climate/weather and power-demand examples where residuals vary smoothly across space, time, or covariates, but clearly fail to be globally stationary or exchangeable.
>      - Emphasize that in operator-learning settings, the residual distribution changes with the input function in ways that are structured (e.g., seasonal cycles, spatial heterogeneity, different dynamical regimes), but often locally smooth in a suitable feature space.
>      - Add an early figure, based on the ERCOT energy demand data, that visually illustrates these kinds of residual patterns and why global exchangeability is unrealistic, whereas local neighborhoods in a feature space are much more plausible.
>
>  __W2__:  The depth explanation in Section 3 has been substantially augmented. We now highlight that the $\Phi$-depth acts as a worst-case projection score, making it automatically sensitive to anisotropy and directional variability in the residuals. Scalar norm-based scores, by contrast, collapse all directions into a single radial measure. This explains the empirical pattern we observe, in both the synthetic and real datasets, that when residuals have elongated or structured geometry, depth-based scores adapt to the dominant modes of variation and achieve noticeably smaller interval scores. By comparison, standard scores need to inflate in all directions, leading to highly inefficient prediction sets. The revised text makes this connection explicit.
>
> __W3__: In the revised Section 4 we include experiments in Figure 2 that show that marginal coverage stays close to nominal levels under locally exchangeable residuals. Previously we were relying on Table 1 to illustrate nominal coverage under local exchangeability via the FC scores being controlled around 0.9 However, the new experiments in Figure 2 now provide the full battery of comparisons with respect to all of the localizer settings ($\lambda$, $H$, $\Phi$, $M$) just as we had done for global exchangeability.
>
> __W4__: We expanded the background section on conformal prediction to include a proper discussion of local exchangeability as an assumption and how it modifies the usual exchangeability setting. We also provided a brief background on finite dimensional conformal and existing finite-dimensional local conformal prediction methods. This section, along with the introduction, has been almost completely rewritten to better introduce the conformal setting and introduce readers to the concept of local exchangeability.
>
> __W5__: The Introduction has been significantly restructured and almost completely rewritten to focus primarily on the problem of distribution-free UQ for function-valued predictors under nonstationary residuals. We almost entirely cut the discussion on neural operators and now only briefly introduce them as a tool for making function valued predictions. We agree that our first iteration of the introduction did not provide a very compelling or sufficient motivation to the problem we were actually solving.
>
> __Q1__: Yes, although the paper focuses on functional settings, several of our experiments operate in effectively low-dimensional regimes. For example, lower-resolution operators such as the energy demand data and air quality data with 24 sample points per function. In these cases, LSCI consistently outperforms global conformal baselines in interval score while maintaining coverage. The mechanism is the same as in higher-dimensional settings: depth-based scores adapt to anisotropic residual geometry, while scalar scores do not. These results suggest that the advantages of depth-based scoring extend naturally to finite-dimensional problems, even though we do not include a dedicated finite-dimensional benchmark.

---

### Official Review · Reviewer_gyDG · 2025-11-01

**Soundness:** 4
**Presentation:** 4
**Contribution:** 4
**Rating:** 8
**Confidence:** 3

**Summary:**

The authors introduce LSCI, a novel, distribution-free UQ framework for operator models, such as NOs. LSCI provides statistically rigorous, function-valued prediction sets that are locally adaptive. The authors propose local Phi scores based on Phi-depth, allowing the method to measure the centrality or typicality of a residual function relative to a local distribution of residuals, rather than just using a single scalar value. LSCI computes a local, test-specific quantile by weighting calibration samples based on their feature-space similarity to the test input.

**Strengths:**

1. The paper is technically sound
2. The paper has strong empirical evidence and is convincing. The experiments show it produces prediction bands that are appropriately tight in low-variance regions and wider in high-variance regions, leading to more informative and useful uncertainty estimates.
3. The theory relies on local exchangeability, a far more realistic assumption for complex, non-stationary data than the standard global exchangeability required by standard CP methods.
4. The authors provide a practical algorithm

**Weaknesses:**

1. The method's adaptivity hinges on the choice of the localization kernel H, the bandwidth lambda, and potentially a feature map phi. While the paper ablates these (in fig 1) and suggests tuning lambda, it offers little guidance on how to choose H or other hyperparameters and analyzes how it affects the resulting efficiency. A discussion on how to choose these parameters would be beneficial.
2. While weaker than global exchangeability, the assumption that residual distributions vary smoothly could be violated in scenarios with abrupt shifts or phase transitions. The paper does not test the method's robustness to such sharp breaks in the data-generating process. A discussion on the failure modes would be beneficial.

**Questions:**

1. Table 4 shows that coverage is robust to the choice of projection Phi. But how does this choice affect the tightness and shape of the prediction bands?
2. Can you provide more intuition or formal guidance on how to select the similarity kernel H and feature map for a new problem?
3. Why were Bayesian Neural Operators (BNOs) or other probabilistic operator models not included as baselines? While they are not distribution-free, they are a primary competing approach for UQ in this domain.
4. How does the method perform if the calibration set is very large? Does the need to compute n local scores for each test point become a practical bottleneck?
5. How does LSCI's coverage and tightness behave if the model is significantly mis-specified or poorly trained (i.e., the residuals are very large and structured)?

---

> ### Author Response · Authors · 2025-12-04
> **Rebuttal by authors**
>
> We thank you for your thoughtful comments and questions. Below we respond to each point and reference the corresponding revisions made in the current manuscript.
>
> __W1__: We expanded Section 3.3 to provide clear, practical guidance on how the bandwidth $\lambda$ controls the locality–stability tradeoff. The revised text explains that small $\lambda$ effectively recovers a global method while larger $\lambda$ concentrates the score distribution around the test input, improving adaptivity but increasing variance. We also now state explicitly how we tune $\lambda$ in practice using a held-out fold and interval score. In Section 4, we added recommendations for choosing $H$ and $\phi$ based on the structure of the residuals. For high-dimensional or spatially structured functions, we advise using lower-dimensional feature maps such as FPCA or Fourier modes. For smoother or low-dimensional settings, the identity embedding is sufficient. We also note that coverage is empirically stable across all choices (as shown in Figure 2 and Tables 2 and 4), and that the primary effect of these hyperparameters is on efficiency. As with $\lambda$, cross validation can be used to find the hyparameter set minimizing the IS or other criteria.
>
> __W2__: You're absolutely right that local exchangeability can, and likely will, be violated during abrupt phase transitions or changepoints. We added a new discussion of this limitation in the Discussion section (lines 535–541), noting that abrupt or discontinuous changes can cause oversmoothing during around the changepoint, which leads to reduced adaptivity. We think a systematic study of locally exchangeable conformal inference under abrupt regime changes is very interesting topic for future research.
>
> __Q1__: Section 3.1 now contains an expanded explanation of how projection families influence the score. The revision clarifies that the $\Phi$-depth acts as a worst-case centrality measure over linear probes, making it sensitive to anisotropy and direction-dependent variability in the residual cloud. In Section 4.1 we emphasize that although marginal coverage is stable across different projection families, interval score and band geometry behave as expected: projection classes aligned with the dominant modes of variation (e.g., Fourier or FPCA on structured fields) yield the most precise prediction sets.
>
> __Q2__: Beyond the hyperparameter discussion above, we have also updated Section 4 to include a simple, rule-of-thumb selection strategy. We recommend tuning $(H, \lambda)$ via interval score on a held-out fold and choosing $\phi$ to stabilize distances: FPCA/Fourier features for high-resolution outputs, and identity features when the outputs are smooth and low-dimensional. The empirical results show that LSCI remains robust across these choices, with variations affecting efficiency but not coverage. The most consistent pairing, which we recommend as a starting point, were the fourier feature maps with $L_\infty$-norm localization.
>
> __Q3__: We added an explicit explanation in Section 4 clarifying that BNO-style methods require prior assumptions and do not provide distribution-free, finite-sample coverage. This actually makes it quite difficult to fairly compare between methods because the interval score can be tuned down by simply having the model undercover the target (to an extent). Thus, to make things fair, we needed to require that all models cover at or around the same level, which is not something probabilistic methods can guarantee in general. Because our focus is on distribution-free uncertainty quantification, we chose baselines that are directly comparable in this regard, such as conformalized DeepONet and UQNO variants.
>
> __Q4__: The revised manuscript now includes a discussion of computational scaling. Evaluating localized scores scales linearly with the calibration set size and the number of depth samples. We explain that for very large calibration sets, one can restrict kernel evaluation to approximate nearest neighbors in feature space without altering the validity guarantees. Although this would incur a space cost if we need to keep track of a nearest neighbors graph. For the sample sizes we considered, however, $n \approx 10^3-10^4$, computing the weights was extremely fast in practice (a few milliseconds).
>
> __Q5__:  We clarified in the theory section that conformal validity depends on (local) exchangeability rather than model correctness. To illustrate the empirical behavior, Figure 3 shows that LSCI’s interval scores, which are closely tied to coverage and tightness, grow much more slowly than those of baselines under fixed bias, input-dependent bias, and covariate shift. This is because the localized score can adjust its location (capturing bias) and shape (capturing structured residuals) to accommodate residual structure efficiently. Marginal coverage remains stable so long as local neighborhoods of residuals remain comparable.

---

### Author Response · Authors · 2025-12-02
**Summary of Revisions and General Response**

We thank all reviewers and the AC for their thoughtful and constructive feedback. We have made substantial revisions to improve clarity, strengthen motivation, and expand both theoretical and empirical explanations. Below we summarize the key updates.

1. __Motivation and Local Exchangeability__:
Several reviewers requested clearer motivation for when global exchangeability fails and when local exchangeability is appropriate. We expanded the Introduction and Section 2 with concrete examples from operator learning, climate and weather prediction, and power systems, where residual distributions vary smoothly but are not globally stationary. __Figure 1__ now visually illustrates this scenario using the energy demand data considered in Section 4. We hope these additions make the problem setting more accessible to both conformal prediction and operator-learning audiences.

2.  __Conformity Score Localization vs. Weighted Quantiles__:
We clarified that LSCI introduces locality through the __conformity score__ rather than through a weighted quantile. This was done because the standard approach assumes a global _score geometry_ (e.g. spherical) and inflates the radius up our down, while our depth based approach allows for locally adaptive geometry in the prediction sets by localizing the score directly. Sections 3.1–3.3 now explicitly distinguish our approach from existing localized CP frameworks (Guan 2023; Hore & Barber 2023; Barber et. al. 2023) and explain why the standard conformal quantile remains appropriate under local exchangeability. We believe these clarifications help address concerns regarding the quantile definition and its relationship to prior work.

3. __Theoretical Guarantees__:
We have corrected the theoretical bounds to strictly match the final algorithm. As pointed out by Reviewer 3, the previous draft referenced a bound for quantile localized conformal procedures. Because LSCI localizes the score but not the quantile, we have removed the extraneous bound and replaced it with the correct coverage-gap bound corresponding to the score-based localization used by LSCI. This result is still reliant on (Barber et. al. 2023), but now, correctly, uses equal weights when bounding the coverage gap and only makes use of their local exchangeability-like condition on the gap. A full, self-contained proof of the corrected __Proposition 3.1__ is now provided in __Appendix A.3__.

4. __Hyperparameter Guidance and Depth Intuition__:
We added a subsection on bandwidth selection and clarified the roles of the localization kernel $H$, feature map $\varphi$, and projection family, $\Phi$. Section 3.1 now provides geometric intuition for depth-based scores, explaining how they adapt to local anisotropy and variability in the residual cloud. We believe this addresses questions regarding tuning and why depth-based scores yield tighter intervals since they do not over-inflate prediction sets in directions of low variability.

5. __Robustness Analysis and Empirical Coverage__:
Section 4 now includes explicit coverage evaluations across localization settings for both exchangeable and locally exchangeable data in __Figure 2__. These results, which were only implicit in __Table 1__, now show that the coverage gap derived in __Proposition 3.1__ is small in both cases, and demonstrate that LSCI maintains stable coverage while improving efficiency (via the baseline comparisons and IS scores). We hope this expanded empirical section addresses reviewer questions coverage under local exchangeability.

6. __Evaluation Metrics and Sampling Procedure__:
To help readers unfamiliar with operator learning interpret the results, we expanded the explanation of the evaluation metrics (IS, FC, BW, CR) and their relevance for function-valued predictions. Section 3.4 clarifies that FPCA sampling is used only for visualization and estimating prediction bands; FC (functional coverage) is computed analytically from the depth-defined prediction set. We believe this resolves concerns regarding the evaluation of implicit functional sets.

7. __General Improvements in Readability and Structure__:
We made several improvements throughout the paper, cleaner notation, expanded explanations in Sections 2–4, and additional details in the appendix. These changes aim to improve readability, particularly for readers less familiar with neural operators or functional regression. In particular, we significantly reduced the discussion on neural operators and now only use them as a motivating example.

Overall, we believe that the revised manuscript addresses the reviewers’ questions and concerns while improving clarity and completeness. We are grateful for the reviewers’ feedback and hope the final version reflects these improvements. We will provide separate, point-by-point responses to each reviewer in subsequent comments below.

---

### Meta-Review · Area_Chair_3XqD · 2026-01-03

**Summary:**

This work introduces a method called Local Sliced Conformal Inference (LSCI), generating function-valued, locally adaptive prediction sets for operator models. It provides finite-sample validity and an upper bound on the coverage gap with experiments validated.
The reviewers’ concerns centered less on the correctness of the proposed method and more on its readability, contributions, scope, robustness, and practical generality, which collectively informed my suggested decision.

Multiple concerns remained that affect confidence in the method’s range of applicability. Reviewers repeatedly noted that the core assumption of local exchangeability, while weaker than global exchangeability, may fail in practically important regimes involving abrupt changepoints or non-smooth residual dynamics. These failure modes were discussed qualitatively in the revision but not empirically explored or mitigated.

**Reviewer Concerns:**

Addressed:

- Motivation for local exchangeability
- Score localization vs. quantile localization
- Issues on hyperparameter
- Intuition or theoretical explanation for proposed methods' advantages.

Reviewer concerns still outstanding or only partially addressed:

- Failure under abrupt regime shifts

**Reviewer Scores:**

gyDG: No change

ngnT: +2

8P52: +2

NpHH: No change

---

### Decision · Program_Chairs · 2026-01-26

Reject